# Reinforcement Learning with State Observation Costs in Action-Contingent Noiselessly Observable Markov Decision Processes

**HyunJi Alex Nam***
Department of Computer Science
Stanford University
`hjnam@cs.stanford.edu`

**Scott L. Fleming***
Department of Biomedical Data Science
Stanford University
`scott.fleming@cs.stanford.edu`

**Emma Brunskill**
Department of Computer Science
Stanford University
`ebrun@cs.stanford.edu`

## Abstract

Many real-world problems that require making optimal sequences of decisions under uncertainty involve costs when the agent wishes to obtain information about its environment. We design and analyze algorithms for reinforcement learning (RL) in Action-Contingent Noiselessly Observable MDPs (ACNO-MDPs), a special class of POMDPs in which the agent can choose to either (1) fully observe the state at a cost and then act; or (2) act without any immediate observation information, relying on past observations to infer the underlying state. ACNO-MDPs arise frequently in important real-world application domains like healthcare, in which clinicians must balance the value of information gleaned from medical tests (e.g., blood-based biomarkers) with the costs of gathering that information (e.g., the costs of labor and materials required to administer such tests). We develop a Probably Approximately Correct (PAC) RL algorithm for tabular ACNO-MDPs that provides substantially tighter bounds compared to generic POMDP-RL algorithms, on the total number of episodes exhibiting worse than near-optimal performance. For continuous-state ACNO-MDPs, we propose a novel method of incorporating observation information that, when coupled with modern RL algorithms, yields significantly faster learning compared to other POMDP-RL algorithms in several simulated environments. []

## 1 Introduction

Reinforcement learning (RL), which involves learning to make sequences of good decisions in stochastic environments, has had many impressive successes over the past decade [60, 68]. While many of the more visible successes have occurred in simulated environments, such as Atari video games [44, 58], reinforcement learning has increasingly been used to solve real-world problems [10, 46, 36, 41, 75, 47, 37, 51, 42, 34]. One key difference between reinforcement learning in simulated vs. real-world environments is that, in most simulated environments, the agent can fully observe the underlying state at each time step with no additional observation costs. In contrast, real-world environments frequently entail state observations that are noisy and/or costly, but optional. Environ-

---

*Equal contribution

35th Conference on Neural Information Processing Systems (NeurIPS 2021).

ments in which state information at each time step can be noisy or missing are more appropriately modeled as partially observable Markov decision processes (POMDPs) rather than MDPs [29].

In this paper, we provide theory and algorithms for a special class of POMDPs in which state information is complete/noiseless when observed, but may be missing at any given time step if the agent chooses not to observe the state. We call this class of POMDPs *Action-Contingent Noiselessly Observable MDPs* (ACNO-MDPs), which can be useful for capturing a number of important real-world settings, such as:

**ACNO-MDPs in healthcare**. Clinicians in Intensive Care Units (ICUs) frequently have to make sequences of treatment decisions under uncertainty for patients at risk. While accurate laboratory tests can inform such decisions, administration of these tests carry a significant cost for the patient and the health system [25]. These costs, together with the fact that frequent testing may be redundant and wasteful [12, 78], appropriately lead clinicians to refrain from constantly observing the patient state (i.e., ordering laboratory tests) [1, 17]. Similarly, other settings – such as glucose monitoring to assist in insulin dosing recommendations [18], or white blood cell count monitoring to assist in anti-HIV drug dosing [15] – have modern tools for accurate observations of biomarkers and could be appropriately modeled by our ACNO-MDP framework.

**ACNO-MDPs for user-adaptive experiences.** Applications on mobile phones and other personal devices can collect information on the user's status, such as location, motion, inter-user contact, and background noise, in order to adaptively suggest features that enhance the user's experience [69, 70]. While these sensors can in theory always be kept active, battery consumption due to constant sensing would make such applications less desirable to the user. Modeling this problem as an ACNO-MDP could enable a policy that balances battery usage with personalizing the user's experience. [1]

**Our contributions.** Our main contributions are:

- Proposing a Probably Approximately Correct "Observe-then-Plan" algorithm for tabular ACNO-MDPs that fully observes while exploring, then employs POMDP planning using learned models. The resulting policy can select when to observe to achieve high expected reward in environments with state observations that are costly but optional.

- Providing a finite sample guarantee for the above algorithm in environments with discrete states and actions, such that the number of episodes exhibiting worse than near-optimal behavior is no larger than $O(\frac{S^5 A^2 H^5 R_{max}^3 \iota^4}{\epsilon^3})$ with $\iota = O(\ln(SAH/\delta\epsilon))$ with high probability. This improves over prior work targeted towards more general POMDP problems (Table 4).

- Developing an algorithm that extends these ideas to ACNO-MDPs with continuous state and action spaces. We introduce a novel belief state encoder that can be coupled with many modern deep RL algorithms to solve problems with observation costs, which we demonstrate with several simulation experiments.

## 2 Related work

Learning in POMDPs without a known model of the environment (i.e., with neither a model of state-to-state transition dynamics nor a model of observations conditioned on states) is significantly more difficult than learning in MDPs [49]. Thus while there is a rich literature on planning in POMDPs with a known environment model [3, 7, 13, 23, 35, 39, 40, 50, 53, 54, 57, 59, 62, 67, 77], including in Mixed Observability MDPs (MOMDPs) [61], active perception frameworks [19, 63], and Non-Observable MDPs (NOMDPs) [38], fewer approaches exist for RL in POMDPs (POMDP-RL). One key difficulty in general POMDP-RL is that, without a known observation model, the agent can never know with certainty which states were actually visited. Approximating a transition model for use in subsequent planning thus becomes infeasible without additional assumptions on the observation model. Many approaches in POMDP-RL, such as the Bayes-Adaptive POMDP (BA-POMDP) [52, 56] and its variants [31, 32], assume access to an informative prior distribution over the observation model and maintain posterior distributions over the environment model parameters. These algorithms plan using belief distributions over states rather than the states themselves. More recent

---

[1]While some of the POMDP-RL methods proposed in this paper may not be energy efficient, recent approaches for compiling and compressing policies into approximately equivalent finite-state controllers could enable energy efficient user-adaptive applications that both sense and change behaviors adaptively [20].

approaches like Deep Recurrent Q-Networks (DRQN) [22, 79] and Deep Variational Reinforcement Learning (DVRL) [43] use neural network-based sequence models to learn a condensed representation of the agent's action and observation history and apply modern RL algorithms (e.g., DQN [44], A2C [72]) on this learned representation. These approaches show promising empirical performance but lack theoretical guarantees and do not leverage the specific structure of ACNO-MDPs.

Theoretical guarantees on a POMDP-RL algorithm's performance can be beneficial when gathering samples under a suboptimal policy is costly. Though recent works have been able to provide theoretical guarantees in surprisingly broad classes of POMDPs, these works either provide regret guarantees (not sample complexity guarantees) [24, 66], or make assumptions about the structure of the observation matrix which do not hold in ACNO-MDPs (e.g., that the observation matrix is full rank [21, 2] or its singular value is bounded away from 0 [27]). To the best of our knowledge, our work therefore provides sample complexity guarantees for a subclass of POMDPs in which no such prior guarantees existed.

Closely related to our work, Klima et al. [33] consider RL in environments where the state is fully observed, but only intermittently. However, whereas Klima et al. [33] consider environments in which the agent cannot control state observation frequency, we address RL in environments where the agent can *decide* when to observe, albeit at a cost. Similarly, Yoon et al. [74] and Chang et al. [8] use RL methods to learn policies for active sensing/dynamic measurement scheduling. In these works, however, the reward is directly tied to prediction accuracy and agents have no influence on the underlying state beyond observation whereas we focus on agents that can influence the state.

Since acceptance, it has come to our attention that earlier work by Zubek and Dietterich [80], Zubek et al. [81], and Bellinger et al. [4] proposed RL approaches for learning optimal policies that both modify and observe the state at each time step, as in our problem setup. However, these prior works only provide algorithms for tabular environments and do not provide any theoretical analysis bounding the sample complexity. Related contemporaneous works by Yin et al. [73] and Bellinger et al. [5] tackle the problem of RL with observation costs in continuous-state POMDPs and MDPs, respectively: Yin et al. [73] considers a setting where individual elements of the state vector can be observed at each point in time and Bellinger et al. [5] performs a more in-depth qualitative analysis of agents' measurement patterns throughout training.

## 3 Problem setting

**Action-contingent noiselessly observable Markov decision processes (ACNO-MDPs).** A discounted, finite-horizon ACNO-MDP is an environment defined by a tuple $\mathcal{M} = \langle \mathcal{S}, \mathcal{A}, \mathcal{O}, p, \gamma, r, b, H \rangle$, where $\mathcal{S}$ and $\mathcal{A}$ are state and action spaces with cardinalities $S$ and $A$, respectively. The action space $\mathcal{A}$ consists of tuples {control actions} $\times$ {observe, not observe}. As in POMDPs, the observation space, $\mathcal{O}$, has cardinality $O$, and is related to $\mathcal{S}$ by the observation/emission function $p(o|s'a)$, which defines the probability of observing $o$ if the agent has taken action $a$ and is now in $s'$. In ACNO-MDPs, however, the observation space is specifically constrained to be $\mathcal{O} = \mathcal{S} \cup \{\text{missingness}\}$, where the "missingness" observation provides no information whatsoever about the underlying state. The "action-contingent noiselessly observable" aspect of ACNO-MDPs derives from the fact that $p(o|s', a_{\text{observe}}) = 1$ if and only if $o = s'$, while $p(o'|s', a_{\text{observe}}) = 0$ for all other observations $o' \neq s'$. Similarly, $p(o|s', a_{\text{not observe}}) = 1$ if and only if $o = $ "missingness".

While observations in ACNO-MDPs are deterministic, transition dynamics may be stochastic. The probability of transitioning to state $s'$ after taking action $a$ from state $s$ is given by $p(s'|s, a)$. We let $b$ represent a belief distribution over possible states, subscript $t$ the time step within the episode, and $H$ the episode length. We denote the number of times the state-action pair $(s, a)$ is visited across episodes as $n(s, a)$.

At each time step, the agent receives expected reward $r(s, a)$, which includes the control-associated reward from executing action $a$ in state $s$ plus an observation cost $c(s, a) \leq 0$. If the state is not observed, $c(s, a) = 0$. Thus, for any action $a$, $r(s, a_{\text{observe}}) = r(s, a_{\text{not observe}}) + c(s, a_{\text{observe}})$. Observation costs are assumed to be known in advance. Without loss of generality, we assume $0 \leq r(s, a) \leq R_{\max}$ for all states $s$ and actions $a$. We assume stationary dynamics, observations, and rewards, but allow for nonstationary policies in light of the fixed horizon length. A $t$-step policy, $\pi_t(\cdot)$, is a function that maps belief state distributions to distributions over actions. We let $V^{\pi_t}(b_0) = \mathbb{E}_{(s_i, a_i) \sim \pi_t} \sum_{i=0}^{t-1} \gamma^i \sum_{s \in \mathcal{S}} b_i(s) r(s, a_i)$ represent the expected return obtained from

starting in $b_0$ and following $t$-step policy, $\pi_t$, with discount factor $\gamma \in [0, 1]$. An optimal policy, $\pi^*$, is a policy for which $V^{\pi^*}(b) = \max_\pi V^\pi(b)$ for all beliefs $b$ in the belief space. The goal of ACNO-MDP learning is to find the policy $\pi$ that maximizes $V^\pi(b_0)$. We evaluate our algorithms using the notion of sample complexity [30, 65], a measure of how much data is needed to learn a near-optimal policy. An algorithm is Probably Approximately Correct (PAC) if its sample complexity is polynomial in relevant quantities (e.g., $S$, $A$, $H$) with high probability [65].

Leveraging the specific observation structure of ACNO-MDPs can improve data efficiency in two ways: (1) when the agent observes during exploration, it can more rapidly learn an accurate approximate transition model; and (2) this improved transition model can be used for better state estimation during POMDP planning, as the underlying state transitions for observed and unobserved states are identical. Faster convergence to an accurate model of the environment and subsequently better planning reduce sample complexity bounds, as we demonstrate both theoretically and empirically.

## 4 Algorithms

We propose two high-level frameworks for finding a near-optimal policy in ACNO-MDP settings. The first is **"Observe before planning"** (Algorithm 1), which initially spends a fixed number of timesteps always observing states in order to learn accurate models of the transition dynamics and rewards, as if in a MDP, then switches to computing a near-optimal POMDP policy using these models. The second is **"Observe while planning"**, which incorporates the known ACNO-MDP observation structure (i.e., that observation emissions are identical to underlying states following a choice to observe) into an existing POMDP-RL algorithm.

Our proposal for "Observe before planning" follows the structure of EEPORL [21], which devotes a fixed number of episodes to exploration in order to accurately estimate models of transition dynamics, rewards, and the function mapping states to observations. We highlight three key differences between EEPORL and our approach: (1) our "Observe before planning" framework only needs to learn models of the transition dynamics and reward function, since the observation model is known and every transition can be exactly identified if the agent chooses to observe; (2) our method can track exactly which state-action pairs are observed during the initial full observability-episodes in which the setting is effectively an MDP; and (3) our method does not require EEPORL's assumptions about state transition probabilities, observations, and rewards (their Assumptions 2 & 3), whereas EEPORL depends on these to guarantee that the outcomes of every action are uniquely identified.

The motivation for "Observe while planning" is that any POMDP-RL algorithm can be simplified in ACNO-MDP settings by leveraging the known ANCO-MDP observation function, which gives the true underlying state if the agent chooses to observe, and an uninformative state otherwise.

The purpose of both frameworks is to improve sample efficiency by leveraging the known observation structure in ACNO-MDP settings, empirically and/or theoretically.

---

**Algorithm 1** Observe before planning
---
1: Set dataset $D \leftarrow \emptyset$
2: **for** episode = 1 to N **do** ▷ N is # of episodes spent exploring (e.g., Thm1 gives one example of N)
3:     Reset to initial state $s_0$
4:     **for** step = 1 to H **do**
5:         Pay to observe every $(s, a, s', r)$ and store into $D$
6:     **end for**
7: **end for**
8: Calculate maximum likelihood estimates of transitions $\hat{p}(s, a, s')$ and rewards $\hat{r}(s, a)$ using $D$
9: Return policy $\hat{\pi}$ from POMDP planning under $\hat{p}, \hat{r}$

---

In our experiments, we instantiate these frameworks into specific algorithms and compare them against MDP-RL methods, which learn an always-observing policy regardless of the cost of observation, as well as state-of-the-art POMDP-RL algorithms (e.g., DRQN [22] and DVRL [43]), which do not leverage any knowledge about the ACNO-MDP observation function. A summary of all algorithms used for comparison is shown in Table 1 and training details are provided in the Appendix.

Table 1: Summary of our algorithms and baseline approaches.

| Framework | Algorithm name | Setting |
|---|---|---|
| Observe before planning | *Observe-then-Plan* (Ours) | Tabular |
| Observe while planning | ACNO-POMCP (Ours) | Tabular |
| Generic POMDP-RL | DRQN [22] | Tabular |
| MDP-RL | EULER-VI [76] | Tabular |
| Observe while planning | ACNO-A2C (Ours) | Continuous States/Actions |
| Generic POMDP-RL | DVRL [43] | Continuous States/Actions |
| MDP-RL | A2C [45, 72] | Continuous states/Actions |

### 4.1 *Observe-then-Plan* algorithm for tabular ACNO-MDPs ("Observe before Planning")

For tabular ACNO-MDP domains, we provide the *Observe-then-Plan* algorithm as an instantiation of the "Observe before planning" framework. *Observe-then-Plan* decouples exploration from planning, following prior work on EEPORL [21] and reward-free exploration [28]. In contrast to this prior work, which addresses non-stationary domains, we focus on domains with stationary dynamics and rewards in a finite episodic setting. The first key algorithmic step in *Observe-then-Plan* is to visit all relevant state-action pairs sufficiently many times, while observing at every time step, so as to learn accurate models of the transition dynamics and rewards. For the purposes of our theoretical analysis, we employ EULER [76] in this exploration phase, as it provides state-of-the-art algorithm cumulative regret guarantees for episodic MDPs. We note, however, that any MDP-RL algorithm can be used for this step. The second key algorithmic step in *Observe-then-Plan* is to use the models of transitions and rewards estimated from exploration data for POMDP planning. While in our analysis we leverage algorithms that guarantee $\epsilon$-optimal performance in POMDP planning [56, 16], these approaches quickly become computationally challenging or intractable for any reasonably-sized POMDP [49]. In our experiments we therefore use POMCP, a popular and fast POMDP planning algorithm [59].

### 4.2 ACNO-POMCP/ACNO-A2C algorithms ("Observe while Planning")

Although our theoretical guarantee (Theorem 1) requires the agent to initially spend a fixed number of episodes exploring while observing at every time step, this extensive exploration and observation may be unnecessary for obtaining near-optimal performance in practice. This motivates our second framework "Observe while planning," under which we incorporate the known observation structure into an existing state-of-the-art POMDP-RL method.

For tabular settings, we instantiate the "Observe while planning" framework with the ACNO-POMCP algorithm. ACNO-POMCP simultaneously (1) uses online POMCP to plan optimal trajectories based on simulated rollouts with the models learned so far, and (2) updates the approximate reward and transition models used in these simulations every time a new sample $(s, a, s', r)$ is observed from the true environment. Note the observation function does not need to be learned because the algorithm leverages the fact that $p(o|a_{\text{observe}}, s') = 1$ if and only if $o = s'$ and $p(o|a_{\text{not observe}}, s') = 1$ if and only if $o =$ "missingness". Thus model estimation in ACNO-POMCP reduces to MDP model estimation, but the algorithm does not quantify the policy's degree of sub-optimality because when and how often to observe are determined by the online RL agent.

For ACNO-MDPs with continuous state spaces, we instantiate the "Observe while Planning" framework with the ACNO-A2C algorithm by modifying DVRL, a state-of-the-art POMDP-RL algorithm for continuous state space settings. As in the DVRL algorithm, ACNO-A2C relies on (1) a belief encoder model (a Gated Recurrent Unit-Recurrent Neural Network, or GRU-RNN [9]) for encoding histories of past actions, observations, and inferred states; (2) a belief updating model (also a GRU-RNN) that updates the encoded history based on the current observation, inferred state, and prior history; (3) a stochastic transition model, $p_\theta(z_t|h_{t-1}, a_{t-1})$, representing the probability of visiting underlying state $z_t$ given encoded history $h_{t-1}$ and action $a_{t-1}$, which is instantiated as a multivariate normal distribution whose mean and diagonal variance are governed by a neural network; and (4) A2C for policy learning [43].

In our work, we modify the DVRL belief encoder model to incorporate the known observation structure as follows: The original DVRL belief encoder maintains a (weighted) set of belief particles,

each of which represents an encoding of the agent's state and action history based on past observations and actions. When a new state observation is received, each particle is updated with a latent state sampled from a stochastic transition model conditioned on the past history and current observation. Each particle is then assigned a weight proportional to both the likelihood of the agent being in the sampled state and the likelihood of that particle emitting the received observation from that state. In ACNO-MDPs, we split this updating process into two cases. First, if the agent chooses to observe, then there is no uncertainty in the underlying state; every belief particle is updated with the same, observed state and the weights for these particles are distributed uniformly. On the other hand, if the agent chooses not to observe, underlying states are stochastically sampled from a learned transition model, as in DVRL, but in this case the learned transition model does not incorporate information about the observation because it will always be the "missingness" state following a non-observing action. Weights assigned to each particle following a no-observation action are proportional to the probability of reaching the sampled state given the encoded history and action chosen in the previous time step. The belief encoder model is only updated after each observing action, in which case the model loss is the negative log likelihood of visiting the observed state under the transition model.

---

**Algorithm 2** ACNO-A2C belief encoder

---

    **Input** Previous belief set $\hat{b}_{t-1}$, observation $o_t$, action $a_{t-1}$.
    **Output** $\hat{b}_t$, and belief encoder loss $\mathcal{L}_t^{\text{enc}}$.
1: Unpack $w_{t-1}^{1:K}, z_{t-1}^{1:K}, h_{t-1}^{1:K}, \hat{h}_{t-1} \leftarrow \hat{b}_{t-1}$
2: **for** every particle $k = 1, \ldots, K$ **do**
3:     Resample previous context $h_{t-1}^k \sim h_{t-1}^{1:K}$ based on weights
4:     **if** $a_{t-1} = $ observe **then**
5:         $z_t^k = o_t, w_t^k = \frac{1}{K}$
6:         $\mathcal{L}_t^{\text{enc}} \leftarrow \mathcal{L}_t^{\text{enc}} - \log p_\theta(o_t | h_{t-1}^k, a_{t-1})$    ▷ Approx. density, **stochastic transition model**
7:     **else**
8:         Sample $z_t^k \sim p_\theta(z_t^k | h_{t-1}^k, a_{t-1})$.    ▷ Sample from the **stochastic transition model**
9:         $w_t^k \leftarrow p_\theta(z_t^k | h_{t-1}^k, a_{t-1})$    ▷ Estimate density under **stochastic transition model**
10:    **end if**
11:    $h_t^k \leftarrow \text{GRU}(h_{t-1}^k, z_t^k, o_t, a_{t-1})$    ▷ **Belief updating model**
12: **end for**
13: $\hat{h}_t \leftarrow \text{GRU}(\text{Concat}(w_t^k, z_t^k, h_t^k)_{k=1}^K)$    ▷ **Belief encoder model**
14: Pack $\hat{b}_t \leftarrow (w_t^{1:K}, z_t^{1:K}, h_t^{1:K}, \hat{h}_t)$ and include $\mathcal{L}_t^{\text{enc}}$ if $a_{t-1} = $ observe.

---

## 5 Analysis

*Observe-then-Plan* outperforms state-of-the-art PAC POMDP-RL sample complexity bounds (OOM-UCB [26], Table 4) by a factor of $A^2 S O^3$ in ACNO-MDP settings, while increasing these bounds by a factor of $\frac{H R_{\max}^3}{\epsilon}$ and logarithmic quantities. We note, however, that prior state-of-the-art PAC POMDP-RL algorithms make assumptions on the structure of the observation matrix that do not hold in ACNO-MDPs, thus limiting the utility of direct comparisons. Specifically, when an agent chooses not to observe in the tabular ACNO-MDP setting, the observation matrix $\mathbb{O} \in \mathbb{R}^{(S+1) \times S}$ has a single row with values 1 corresponding to the "missingness" observation and all other rows are zero (since each column vector represents $P(o | s', a)$ for a given $(s', a)$ and $a_{\text{not observe}}$ deterministically returns $o = $ "missingness".) This leads to rank deficiency in the observation matrix, violating assumptions made in Guo et al. [21], and $\sigma_{\min}(\mathbb{O}) = 0$, violating the assumptions made in Jin et al. [26]. Thus, to the best of our knowledge, our results provide sample complexity guarantees for a class of POMDPs not covered by prior POMDP-RL work. These guarantees hold for ACNO-MDPs with arbitrary state dynamics and hard-to-reach states.

**Theorem 1.** *Let $\mathcal{M}$ be any ACNO-MDP with observation cost $c(s, a_{\text{observe}}) < 0$ for all $(s, a) \in S \times A$, and reward bounded between $[0, R_{\max}]$. Let $b_0$ be the initial belief, and let $\epsilon$ and $\delta$ be two positive real numbers. Following* Observe-then-Plan *will achieve an expected episodic reward of $V(b_0) \geq V^*(b_0) - \epsilon$ after a number of episodes that is bounded by $O(\frac{S^5 A^2 H^5 R_{\max}^3 \iota^4}{\epsilon^3})$, where $\iota = O(\ln(SAH/\delta\epsilon))$, with probability at least $1 - \delta$.*

Table 2: Sample complexity comparisons

| Algorithm | Setting | Transition | Sample Complexity |
|---|---|---|---|
| OOM-UCB [26] | POMDP | non-stationary | $\tilde{O}\left(\frac{H^4 S^6 A^4 O^3}{\sigma_{\min}(\mathbb{O})^8 \epsilon^2}\right)$ |
| EEPORL [21] | POMDP | stationary | $\tilde{O}\left(\frac{H^4 V_{\max}^2 A^{12} R^4 O^4 S^{12}}{C_{d,d,d}(\frac{\delta}{3}^2)\underline{\sigma}_a(T_a)^6 \underline{\sigma}_a(R_a)^8 \underline{\sigma}_a(O_a)^8 \epsilon^2}\right)$ |
| *Observe-then-Plan* | ACNO-MDP | stationary | $\tilde{O}\left(\frac{H^5 S^5 A^2 R_{\max}^3}{\epsilon^3}\right)$ |

*Proof sketch.* (Full proofs provided in the Appendix). First, we note that prior work on POMDP-RL provides guarantees on expected performance, i.e. $V(b_0) \geq V^*(b_0) - \epsilon$, provided that value function estimates are sufficiently close to the true value function [21]. With sufficiently accurate estimates of the ACNO-MDP transition and rewards models, we can bound the error of the value function estimates computed under those models (following the EEPORL analysis) [16, 21, 55]. Prior work on Model-based Interval Estimation [64, 71] allows us to bound model estimate errors (i.e., $||\hat{p} - p||_1$ and $|\hat{r} - r|$) in terms of the number of visits, $n(s,a)$. Using this, we can find the number of times each state-action pair must be visited in order to guarantee sufficiently accurate ACNO-MDP model estimates. Call this number $m$. Reward-free exploration [28] with EULER [76] guarantees that, with high probability, the agent will visit each state-action pair at least $m$ times after a certain number of episodes, $N$. We show that this number $N$ is $O(\frac{S^5 A^2 H^5 R_{\max}^3 \iota^4}{\epsilon^3})$; in other words, after $N$ episodes of EULER for exploration, the agent has visited sufficiently many state-action pairs, $(s,a)$, to learn a near-optimal policy that trades off between observing and not observing.

Only the exploration phase of *Observe-then-Plan* contributes to its sample complexity, as planning requires no additional sample collection. In generic POMDP settings where the true underlying states are inaccessible, bounding the estimated model errors is difficult because the number of times each state-action pair is visited cannot be measured. In ACNO-MDP settings, transition and reward models can be estimated via simple MLE if the agent is willing to initially pay the cost of observation. Note that in ACNO-MDP settings, each control action has two versions: an observed version and an unobserved version. In both versions, the influence of the action on the state is identical (i.e., $p(s'|s, a_{\text{observe}}) = p(s'|s, a_{\text{not observe}})$), and the cost of observation is assumed to be known. Therefore the learned dynamics and reward models from the observed version of an action can also be used for its unobserved counterpart, and when planning an optimal action sequence, the agent can determine whether the value of deterministically observing the true state, instead of relying on the estimated state transition model, outweighs the cost of observation.

## 6 Experimental results

In order to analyze the empirical performance of our proposed methods in both tabular and continuous state-action spaces, we evaluate the algorithms from Table 1 in (1) a tabular ACNO-MDP environment designed to simulate patient care in the Intensive Care Unit; (2) a continuous-state, discrete-action "Cart Pole" environment; and (3) a continuous-state, continuous-action "Mountain Hike" environment.

### 6.1 Algorithms for tabular ACNO-MDPs

**Sepsis simulator** Our first environment simulates a Sepsis patient and the effects of several common treatments. Adapted from Oberst and Sontag [48], our "Sepsis simulator" defines the state space as {heart rate, blood pressure, oxygen concentration, glucose level}, each of which can take on several discretized state values (so in total, tabular $S = 720$). The action space is defined as a combination of binary treatment options consisting of {antibiotics, vasopressors, ventilation}, each of which can either be "on" or "off". We augment the action space by coupling each action with options of observing and not observing. A reward of 0 is given if the patient reaches a terminal "death" state, 1 if the patient reaches a terminal "discharge" state, and 0.25 otherwise. We provide results for each algorithm under two different fixed observation costs, $-0.1$ and $-0.05$.

**Algorithms** We compare four different methods for learning an optimal policy in the Sepsis simulator. For learning a POMDP policy, the algorithm selects the highest rewarding action from Monte-Carlo rollouts for computational tractability instead of computing the exact policy.

Table 3: Average discounted returns (mean $\pm$ 1 standard error) on Sepsis. Standard errors represent the standard error of the mean return obtained from 3 separate seeds, each with rewards averaged across 50 simulated rollouts at the end of 2000 training episodes using greedy action selection.

| Observation Cost | *Observe-then-Plan* (Observe before planning) | ACNO-POMCP (Observe while planning) | DRQN (Generic POMDP-RL) | EULER-VI (MDP-RL) |
|---|---|---|---|---|
| -0.1 | **0.754 $\pm$ 0.011** | 0.602 $\pm$ 0.029 | 0.593 $\pm$ 0.079 | 0.492 $\pm$ 0.011 |
| -0.05 | **0.740 $\pm$ 0.017** | 0.625 $\pm$ 0.024 | 0.593 $\pm$ 0.036 | 0.638 $\pm$ 0.025 |

*Observe-then-Plan* (instantiation of "Observe before Planning") is our originally proposed *Observe-then-Plan* algorithm (Section 4.1) with a reduced number of episodes (compared to the proposed $H^5 S^5 A^2 / \epsilon^3$ episodes) spent on exploration, in order to make the algorithm computationally tractable.

*ACNO-POMCP* (instantiation of "Observe while Planning") is an adaptation of the POMCP online planning algorithm to the learning setting based on the open-sourced POMCP codebase [14]. At each time step, *ACNO-POMCP* uses Monte Carlo tree search to select actions (including observing and non-observing actions) using an approximated model of the environment and an $\varepsilon$-greedy exploration strategy, with a decaying $\varepsilon$. Whenever *ACNO-POMCP* chooses to observe the state twice in a row, the observed $(s, a, s', r)$ tuple is used to update the parameters of its approximate transition and rewards models. *ACNO-POMCP* treats emitted observations as ground truth state observations and updates its reward and transition estimates over states. Unlike *Observe-then-Plan*, *ACNO-POMCP* does not specify the number of necessary observations to be made prior to planning.

*DRQN* (Generic POMDP-RL) leverages an LSTM layer to encode sequences of one-hot-encoded observations into a single hidden state encoding [22][2]. The algorithm learns a mapping from this hidden state to $Q$ estimates, which are used for action selection under a decaying $\varepsilon$-greedy strategy.

*EULER-VI* (Always observing MDP-RL) chooses to observe at every time step, essentially treating the ACNO-MDP as a MDP with constant observation costs. Initially, EULER-VI uses EULER [76] to strategically explore the state-action space. After a fixed number of "exploration" episodes, EULER-VI computes Value Iteration under maximum likelihood estimates of the transition dynamics and rewards models to learn an optimal five-step nonstationary policy assuming the observation cost is applied at every step.

We evaluate the policy obtained from each method with 50 rollouts in the true environment and report the obtained discounted returns including the observation costs (Table 3).

**Results** We ran each of the above methods for 2000 episodes on the Sepsis simulator domain. The "exploration" phases of the *Observe-then-Plan* and *EULER-VI* algorithms were fixed to be 1000 episodes, at which point learning stopped. *ACNO-POMCP* and *DRQN* learned continuously for all 2000 episodes. Learning curves for each algorithm are shown in Figure 1 and performance of the final policies (i.e., after 2000 episodes) for each algorithm are shown in Table 3.

Figure 1 shows that the algorithms without any explicit "exploration" phase (i.e., *DRQN* and *ACNO-POMCP*) achieved higher returns during the first 1000 episodes compared to algorithms that used the first 1000 episodes to actively explore the state-action space while always observing (i.e., *Observe-then-Plan*, *EULER-VI*). Rewards under both the *Observe-then-Plan* and *EULER-VI* algorithms improved dramatically after switching from "exploration" to "planning", but *EULER-VI* continued to observe at every time step and suffered in performance as a result. We note that, in the Sepsis simulator, any policy that prevents the patient from dying or being discharged for all 5 steps achieves a discounted return of $0.693$. *Observe-then-Plan* is the only algorithm that outperforms this neutral policy in expectation. The gray horizontal line indicates the average returns from POMCP planning with the true model parameters, which suffers from reward approximation errors during planning due to its use of Monte Carlo simulations. Despite the potential error in planning due to stochastic rollouts, we chose POMCP as a computationally tractable POMDP planning algorithm.

---

[2]For implementation, we modified the open-sourced code at `https://github.com/Bigpig4396/PyTorch-Deep-Recurrent-Q-Learning-DRQN`. Our code base is available at `https://github.com/nam630/acno_mdp`

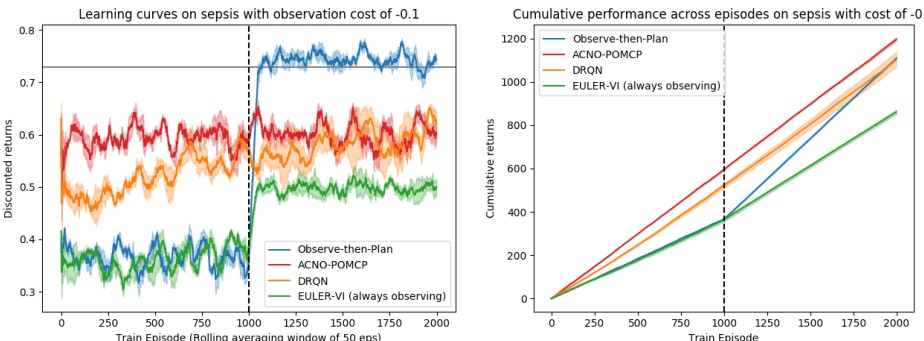

Figure 1: Sepsis learning curves under the observation cost -0.1, with a rolling average window of size 50. Shaded regions indicate $\pm$ 1 standard error over 3 simulated runs. The dotted vertical line at 1000 indicates the episode at which *Observe-then-Plan* and EULER-VI switch to planning. DRQN and ACNO-POMCP continue learning.The $x$-axis shows the number of training episodes and the $y$-axis shows (Left) acquired returns per episode and (Right) cumulative returns across episodes. The gray horizontal line (Left) shows the average returns from POMCP planning with the true model parameters. Note that approximate planning with POMCP, which was chosen over exact POMDP planning for computational tractability, is imperfect even with a perfect model.

## 6.2 Algorithms for ACNO-MDPs with continuous state-action spaces

Explicit reward-free exploration is difficult to do in continuous state and action spaces where the number of times an agent visits a unique state-action pair cannot be measured effectively. Our *ACNO-A2C* algorithm instantiates the second framework of "Observe while Planning" to simultaneously learn an accurate model of the environment while maximizing discounted episode returns. The results suggest that our proposed belief state encoder based on "Observe while Planning" empirically outperforms state-of-the-art algorithms without having to explore all of the state and action spaces.

**Experimental domains**

*Modified CartPole simulator.* We test on OpenAI gym `CartPole-v0` [6] with the following modifications: (1) the action space is doubled in size to {push left, push right} × {observe, not observe}; (2) a cost of -0.4 is applied to every reward $r(s, a_{\text{observe}})$ in which the agent chooses to fully observe the state; and (3) if a state is not observed, then the environment returns all zeros for the observation.

*Mountain Hike simulator.* We also test on a two-dimensional Mountain Hike environment, as described and implemented by Maximilian et al. [43]. The goal of this task is to follow a high reward path along a "mountain ridge". The state space is defined by the current position, $(x, y)$, and the action $a$ is defined as any possible vector $(\triangle x, \triangle y) \in \mathbb{R}^2$ with bounded $\ell_1$-norm. The agent starts from sampled coordinates $(x_0, y_0) \sim N(-8.5 I_{2 \times 2}, I_{2 \times 2})$. Observations returned are exactly the underlying state, except when the agent chooses not to observe in which case observations are a vector of zeros. Transitions are defined as $(x', y') = (x, y) + (\triangle x, \triangle y) + \varepsilon, \varepsilon \sim \mathcal{N}(0, 0.25 I_{2 \times 2})$. The reward is given by $\gamma(x, y) - 0.01 ||a||_1 + c_{obs}(a)$, where $\gamma(x, y)$ is the reward associated with the agent's proximity to the "mountain ridge" and $c_{obs}(a)$ is the cost of fully observing the state, such that $c_{obs}(a_{\text{observe}}) = -0.5$ if the agent chooses to observe and $c_{obs}(a_{\text{not observe}}) = 0$ otherwise.

**Algorithms**

*ACNO-A2C* ("Observe while Planning") is the algorithm instantiating the belief encoder in Algorithm 2 with A2C, a well-known actor-critic reinforcement learning method [45, 72].

*DVRL* (Generic POMDP-RL) uses an implementation of the Deep Variational Reinforcement Learning algorithm by Maximilian et al. [43]. As a state-of-the-art POMDP-RL approach for general POMDPs, this algorithm does not leverage the special observation model given in ACNO-MDPs and must learn both the transition dynamics and the observation model online.

*A2C* (Always observing MDP-RL) applies the actor-critic method [45, 72] directly to the state, assuming the agent always chooses to observe the underlying state and incurs the associated observation

cost. We use the A2C implementation by Maximilian et al. [43] but modified to be defined over states, assuming every action outcome is observed at the fixed cost.

Both the *ACNO-A2C* and *DVRL* algorithms use an encoder that returns an array of aggregated belief particles. This set of belief particles is then used for learning an A2C policy and value function.

**Results** Figure 2 shows the learning curves for our proposed algorithm (ACNO-A2C), the Deep Variational Reinforcement Learning algorithm (DVRL), and the always-observing A2C algorithm. Results are averaged over 5 random seeds. Shaded regions show $\pm$ 1 standard deviations over the 5 runs. Given an equal number of training steps, our ACNO-A2C algorithm achieved a higher-reward final policy, in expectation, and found this high-reward policy in almost half the time that it took for the DVRL algorithm to find its highest-reward policy. Running standard A2C while always observing achieved high returns faster than ACNO-A2C (*Mountain Hike*) or about as fast as ACNO-A2C (`CartPole-v0`), but achieved an inferior final policy relative to ACNO-A2C.

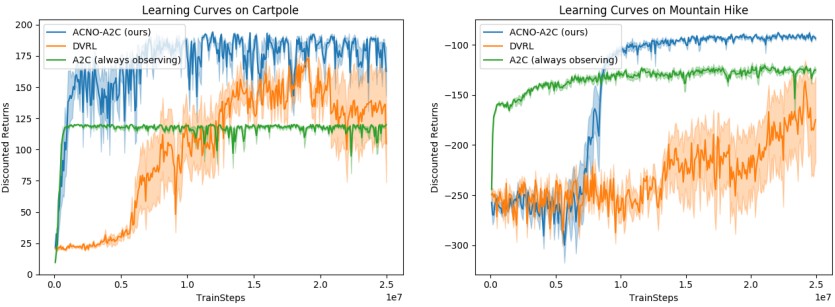

Figure 2: Left: CartPole-v0 with observation cost 0.4. Right: Mountain Hike with cost 0.5.

Our findings suggest that leveraging information about the structure of the observation model can lead to improved learning efficiency (in terms of sample complexity) and higher-reward policies compared to state-of-the-art generic POMDP-RL algorithms which make minimal assumptions about the structure of the observation model.

## 7 Conclusion

We illustrated how assumptions made by many existing PAC POMDP-RL algorithms regarding the observation function are violated in ACNO-MDPs, a special class of POMDPs. We proposed and analysed a PAC RL *Observe-then-Plan* algorithm for efficient learning in such settings. We also proposed a method for encoding belief states in ACNO-MDPs, and a meta-algorithm that can couple this belief representation with deep RL algorithms to learn policies that balance the cost of observing with the benefits of observing for exploration and planning.

We focused on obtaining a sample complexity result under the probably approximately correct (PAC) framework, which has been extensively considered in MDPs, but this approach ignores potentially significant costs incurred during the exploration phase. In contrast, the regret framework incorporates costs incurred at all stages of learning into notions of optimality. One research direction for constraining the cost of strategic exploration would be to develop ACNO-MDP RL algorithms that optimize regret.

We considered the online setting in this work, but offline RL could help mitigate concerns regarding online exploration in safety-critical domains. Exploring extensively to learn dynamics before planning may be infeasible and/or unethical in some settings. For example, in the context of using RL in the ICU to learn optimal policies for Sepsis management, strategic exploration may require the agent to take actions that do not match best practices or existing clinical guidelines; leveraging our ideas to learn optimal policies using data collected from suboptimal behavior policies could help in achieving practical policies with limited online exploration. However, we believe that our work takes a useful step towards characterizing the provable benefits in sample efficiency of ACNO-MDP RL compared to generic tabular POMDP-RL. In generic continuous-state settings, sample complexity analysis and regret analysis for POMDPs or even MDPs remain important open areas of research.

## Acknowledgments and Disclosure of Funding

This work was supported by a NSF CAREER Grant (Emma Brunskill), a National Defense Science and Engineering Graduate Fellowship (Scott Fleming), and a Stanford Graduate Fellowship (Scott Fleming).

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
