Table 4: Sample complexity comparisons with MDP and POMDP-RL algorithms

| Algorithm | Setting | Transition | Sample Complexity |
|---|---|---|---|
| OOM-UCB [26] | POMDP | non-stationary | $\tilde{O}\left(\frac{H^4 S^6 A^4 O^3}{\sigma_{\min}(\mathbb{O})^8 \epsilon^2}\right)$ |
| EEPORL [21] | POMDP | stationary | $\tilde{O}\left(\frac{H^4 V_{\max}^2 A^{12} R^4 O^4 S^{12}}{C_{d,d,d}(\frac{\delta}{3}^2)\underline{\sigma}_a(T_a)^6 \underline{\sigma}_a(R_a)^8 \underline{\sigma}_a(O_a)^8 \epsilon^2}\right)$ |
| *Observe-then-Plan* | ACNO-MDP | stationary | $\tilde{O}\left(\frac{H^5 S^5 A^2 R_{\max}^3}{\epsilon^3}\right)$ |
| Reward-free exploration [28] | MDP | non-stationary | $\tilde{O}\left(\frac{H^5 S^2 A}{\epsilon^2} + \frac{S^4 A H^7}{\epsilon}\right)$ |

# Appendix

## Appendix A. Proofs

The analysis in Appendix A focuses on the sample complexity guarantee of the following instantiation of *Observe-then-Plan* (Algorithm 3) using EULER [75] as a black-box algorithm for reward-free exploration [28]. We note our sample complexity guarantee holds for tabular state and action settings with reward bounded between $[0, R_{\max}]$. As described in Problem Setting, ACNO-MDPs assume $p(o|s', a_{\text{observe}}) = 1$ if and only if $o = s'$ while $p(o'|s', a_{\text{observe}}) = 0$ for all other observations that do not match the true underlying state (i.e., $o' \neq s'$), and $a_{\text{not observe}}$ yields a missing observation with $p(o_{\text{missing}}|s', a_{\text{not observe}}) = 1$. In this section, we show that with a sufficiently large $N_0$, after $SAN_0$ episodes of exploration using EULER, the ACNO-MDP model estimates can yield an $\epsilon$-optimal policy with probability at least $1 - \delta$.

We use a constant $N_{(s_i, a_i)}$ to denote the number of episodes spent on each state-action $(s_i, a_i)$ pair to collect sufficiently many samples. This $N_{(s_i, a_i)}$ provides guarantees on the model estimation accuracy if that $(s_i, a_i)$ pair is $\zeta$-significant (analogous to $\delta$-significant states by Jin et al. [28]; review Definition 4). We also use $n(s, a, s'), n(s, a)$ to represent the counts of samples from the dataset $D$ that are used for MLE estimation (note $N_{(s_i, a_i)} \neq n(s, a)$ since $n(s, a)$ is the number of times EULER successfully makes a visit to the pair during exploration and $N_{(s_i, a_i)}$ is a constant determined by relevant quantities from $\mathcal{M}$ such as $S, A, H, R_{\max}$).

---

**Algorithm 3** Observe-then-Plan

**Input** original environment POMDP $\mathcal{M}, \mathcal{S}, \mathcal{A}, \delta, S, A, N_{(s_i, a_i)}$, obs cost $c(\cdot)$
**Output** policy $\hat{\pi}$ mapping state beliefs to actions
1: **Phase 1:**
2: Set dataset $D \leftarrow \emptyset$
3: **for** all $(s_i, a_i) \in \mathcal{S} \times \mathcal{A}$ **do**
4:     Create a POMDP $\mathcal{M}'$ with state-space $\mathcal{S}$, action-space $\mathcal{A}$ from $\mathcal{M}$
5:     Change rewards in $\mathcal{M}'$ so that $r(s) = \mathbb{1}[s = s_i]$ for the target state $s_i$
6:     Make $s_i$ a terminal state in $\mathcal{M}'$
7:     Fix $\pi(s_i) = a_i$
8:     $\Phi^{(s_i, a_i)}, (s_i, a_i, s', r) \leftarrow \textbf{EULER}(\mathcal{M}', \mathcal{M}, N_{(s_i, a_i)}, \delta' = \frac{\delta}{4SA})$
9:     $D \leftarrow D \cup \{s_i, a_i, s', r\}$ (collect all samples observed while executing **EULER**)
10: **end for**
11: Return the dataset $D$
12: **Phase 2:**
13: **for** all $(s, a, s', r) \in D$ **do**
14:     $\hat{T}(s, a, s') = \frac{n(s, a, s')}{n(s, a)}$
15:     $\hat{R}(s, a) = \frac{\sum r(s, a)}{n(s, a)}$
16: **end for**
17: $\hat{\pi} = \text{POMDP plan}(\hat{T}, \hat{R}, c_{\text{obs}})$
18: Return policy $\hat{\pi}$.

---

Line 6 of Algorithm 3 guarantees that the EULER exploration rewards are bounded between $[0, 1]$ as required since they represent the probabilities of visiting a target state (i.e., the target state cannot be visited multiple times to accumulate reward larger than 1 if the target state is also a terminal state). However, for efficient data collection, once $s_i$ is reached, we could continue to execute $a_i$ to observe the next state $s'$ and collect the tuple of $(s_i, a_i, s', r) \sim \mathcal{M}$ to the dataset $\mathcal{D}$.

We begin our analysis of the above algorithm with several important definitions and comments.

**Sample Complexity.** We evaluate our algorithm using the notion of sample complexity, which measures how much data is needed to learn a near-optimal policy [30, 65]:

**Definition 1** (Sample complexity). *Let $c = (s_1, a_1, r_1, s_2, a_2, r_2, ...)$[3] be a random path generated by executing an algorithm $\mathcal{A}$ for a MDP $\mathcal{M}$. For any fixed $\epsilon > 0$, the sample complexity for exploration (sample complexity, for short) of $\mathcal{A}$ is the number of time steps $t$ such that the policy at time $t$, $\mathcal{A}_t$ satisfies $V^{\mathcal{A}_t}(s_t) < V^*(s_t) - \epsilon$.*

This extends to POMDP settings by replacing $s_t$ with $b_t$ which represents the belief over states. In particular, we show that *Observe-then-Plan* achieves $V(b_0) \geq V^*(b_0) - \epsilon$ with a policy obtained after a fixed number of episodes.

**Probably Approximately Correct (PAC).** A near-optimal policy is defined by this $\epsilon$ suboptimality gap between the algorithm's value estimate and the value estimate under the optimal policy. Under the Probably Approximately Correct (PAC) framework [65], we consider a relaxed notion of sample complexity:

**Definition 2** (Probably Approximately Correct). *An algorithm $\mathcal{A}$ is said to be PAC if, for any $\epsilon > 0$ and $0 < \delta < 1$, the sample complexity of $\mathcal{A}$ is less than some polynomial in the relevant quantities $(S, A, 1/\epsilon, 1/\delta, 1/(1 - \gamma))$, with probability at least $1 - \delta$.*

We represent value function estimates over belief states as the dot product of a belief $b$ and an $\alpha$-vector, $V(b) = \langle b, \alpha \rangle$.

**Definition 3** ($\alpha$-vector equation). *The $\alpha$-vector for a $t$-step conditional policy $\pi_t$ is defined as:*

$$\alpha_t^{\pi_t}(s) = \sum_{a \in \mathcal{A}, o \in \mathcal{O}, s' \in \mathcal{S}} \pi_t(a|s) p(s'|s, a) p(o|s', a) \{r(s, a) + \gamma \alpha_{t-1}^{\pi_{t-1}}(s')\}. \tag{1}$$

We observe that since the observation function $p(o|s', a)$ is 1 for a single $o \in O$ (if $a_{\text{not observe}}$, then $o$ is deterministically a missingness observation and otherwise, $o = s'$ for the true underlying state $s'$) and otherwise 0, we can simplify the $\alpha$-vector for ACNO-MDP settings as:

$$\alpha_t^{\pi_t}(s) = \sum_{a \in \mathcal{A}, s' \in \mathcal{S}} \pi_t(a|s) p(s'|s, a) \{r(s, a) + \gamma \alpha_{t-1}^{\pi_{t-1}}(s')\}. \tag{2}$$

Note that in ACNO-MDP settings, we can view each action as having two versions: an observed or unobserved version. In both versions, the influence of the action on the state is identical: $p(s'|s, a)$ is the same for the unobserved version of a and the observed version. Therefore the learned dynamics from the observed version of an action $a_i$ is identical to its unobserved counterpart. The only difference is that the unobserved action counterpart is associated with deterministically observing the "missingnes" observation instead of deterministically observing the true state.

We now proceed in stating and proving the main theorem of our analysis.

**Theorem 2.** *Let $\mathcal{M}$ be any ACNO-MDP with observation cost $c(s, a_{\text{observe}})$ for all $(s, a) \in \mathcal{S} \times \mathcal{A}$, and reward bounded between $[0, R_{max}]$. Let $b_0$ be the initial belief distribution over states such that $b_0 \in \mathbb{R}^{|\mathcal{S}|}$, and let $\epsilon$ and $\delta$ be two positive real numbers. Following* Observe-then-Plan *will achieve an expected episodic reward of $V(b_0) \geq V^*(b_0) - \epsilon$ after a number of episodes that is bounded by $O(\frac{S^5 A^2 H^5 R_{max}^3 \iota^4}{\epsilon^3})$, where $\iota = O(\ln(SAH/\delta\epsilon))$ with probability at least $1 - \delta$.*

This is stated as Theorem 1 in the main text. A proof sketch is also provided in the main manuscript, though we additionally provide a brief overview of the proof strategy here for clarity. We note

---

[3]Note this defines $c$ as an infinite horizon. We can adapt this definition to the episodic finite horizon case by considering $c$ as the concatenation of state-action sequences from multiple episodes.

from the prior work on POMDP-RL PAC analysis by Guo et al. [21] that with sufficiently accurate estimates of the POMDP transition and reward models, we can bound the value function estimate errors computed under those models. We follow the analysis of EEPORL [21] to bound value function estimates using the $\alpha$-vectors in order to achieve similar results in ACNO-MDP settings. In order to obtain sufficiently accurate ACNO-MDP model estimates, we closely follow the recent work on reward-free exploration by Jin et al. [28], but we assume stationary transition dynamics (unlike the original work which assumes non-stationary transitions).

Below we include relevant lemmas and proofs to show the total number of episodes necessary for obtaining an $\epsilon$-optimal policy in any ACNO-MDP setting by following our proposed tabular algorithm *Observe-then-Plan*. Note that only exploration contributes to this sample complexity since planning does not assume any further updating of the model estimates.

*Proof.*

We use the results from Lemma 1, 2, and 3, which are based on Model-based Interval Estimation [64, 65] and reward-free exploration [28], to compute the necessary number of episodes for obtaining sufficiently accurate ACNO-MDP model estimates with high probability. Once the model estimate errors are guaranteed, Lemma 4 and Lemma 5 show that POMDP planning under these model estimates using approximate $\alpha$-vectors is $\epsilon$-optimal with probability at least $1 - \delta$.

In order to make claims about the probability with which we can achieve $\epsilon$-optimal policies, we must consider the failure events (and their associated probabilities) that could make our algorithm suboptimal even after the specified number of episodes. These failure events are listed as (2.1-4) below. Here $m$ denotes the required number of visits per $(s, a)$ as determined by Lemma 1, $N_{(s,a)}$ is the number of episodes EULER is executed on target $(s, a)$ determined by Lemma 3, $\zeta$ is a fixed value between $[0, 1]$ representing a probability threshold determined by $S, R_{\max}, H$ (review Definition 4), and $\epsilon_R, \epsilon_T$ denote the reward and transition model errors. We call the set of $\zeta$-significant states $K$ and the set of insignificant states $\tilde{K}$.

$$F^{\text{Chernoff}} = \left\{ \exists s \in K, a : \sum_{i=1}^{N_{(s,a)}} 1[X_i] = n(s,a) < m, \text{where } X_i = [(s,a)\text{is executed in the } i^{th} \text{ episode}] \right\}$$

(2.1)

$$F^{\text{EULER}} = \left\{ \exists s \in K, a : \mathbb{E}_{\pi \sim \Phi_{(s,a)}} P^\pi(s, a | s_0) < \frac{\zeta}{2} \right\}$$ (2.2)

$$F^R = \{ \exists s \in K, a : |\hat{r}(s, a) - r(s, a)| > \epsilon_R \}$$ (2.3)

$$F^T = \{ \exists s, \in K, a : ||\hat{p}(s, a) - p(s, a)||_1 > \epsilon_T \}$$ (2.4)

We show that the combined failure probability can still be bounded by $\delta$ so long as our algorithm can guarantee the accuracy of ACNO-MDP transition and reward model parameters (i.e., if we can bound the errors of estimated transition and reward model parameters by $\epsilon_T$ and $\epsilon_R$, respectively).

Lemmas 1 through 3 guarantee that the probability of each failure event for a particular $(s, a)$ pair is bounded by $\delta'$, thus the failure probabilities over all states and actions are also bounded as follows:

$$P(F^{\text{Chernoff}}) \leq SA\delta', P(F^{\text{EULER}}) \leq SA\delta', P(F^R) \leq SA\delta', P(F^T) \leq SA\delta'. \quad (2.5)$$

By union bound, *Observe-then-Plan* has a failure probability of at most $4SA\delta'$. With probability at least $1 - 4SA\delta'$, *Observe-then-Plan* guarantees that the algorithm visits each $\zeta$-significant state-action pair at least $m$ times after a certain number of episodes, $N$ (i.e., $SAN_{(s,a)}$). It is also important to note that the algorithm does not need the same guarantee for $\zeta$-**insignificant** state-action pairs because $\zeta$ is set (Lemma 4) such that even if the transition and reward estimates of the insignificant state-action pairs are far from the truth, they do not affect the $\alpha$-vector estimates too much. By choosing $\zeta = \frac{\epsilon}{18SHR_{\max}}$, $\epsilon_R = \frac{\epsilon}{6SH}$, $\epsilon_T = \frac{\epsilon}{6SH^2 R_{\max}}$ and $\delta' = \frac{\delta}{4SA}$, we show this number $N$ is $O\left(\frac{S^5 A^2 H^5 R_{\max}^3 \iota^4}{\epsilon^3}\right)$ where $\iota = O(\ln(SAH/\epsilon\delta))$. This $N$ gives the final sample complexity of $\widetilde{O}\left(\frac{S^5 A^2 H^5 R_{\max}^3}{\epsilon^3}\right)$ for obtaining an $\epsilon$-optimal policy in any ACNO-MDP setting by following *Observe-then-Plan*. $\square$

We first formally define $\zeta$-significant states based on how likely the states are to be visited under any policy. This definition is based on the definition of $\delta$-significant states by Jin et al. [28]. $\zeta$-significant states are used during the exploration phase of *Observe-then-Plan* to show that only the $\zeta$-significant state-action pairs (whose probabilities of being seen under some possible policy exceed a fixed threshold value) need to be visited sufficiently many times in order to guarantee an $\epsilon$-optimal ACNO-MDP policy with high probability. We define $\zeta$-significant states as,

**Definition 4.** *A state $s$ is $\zeta$-significant if there exists a policy $\pi$ such that the probability of reaching $s$ within $H$ steps from the initial state $s_0$ following $\pi$ is at least $\zeta$. Any action taken from $s$ makes a $\zeta$-significant state-action pair. Formally, $s$ is $\zeta$-significant iff there is some policy $\pi$ such that,*

$$\max_\pi \sum_{t=0}^{H-1} P^\pi(s_t = s|s_0) \geq \zeta. \tag{1.1}$$

For notational simplicity in Lemma 2 and Lemma 4, we use $\max_\pi P^\pi(s)$ to denote $\max_\pi \sum_{t=0}^{H-1} P^\pi(s_t = s|s_0)$ (i.e., the cumulative probability of visiting state $s$ within $H$ steps from the initial state $s_0$). Using this definition of $\zeta$-significant state-action pairs, we now determine the necessary number of visits to every $\zeta$-significant state-action pair in order to bound the ACNO-MDP model estimates based on the Model-based Interval Estimation [64, 65].

**Lemma 1.** *For any state-action pair $(s, a)$, let $\hat{r}(s, a)$ be the sample mean of the observed rewards, $\hat{p}(s, a)$ the empirical transition probability distribution, and $n(s, a)$ be the number of times $(s, a)$ is visited in the ACNO-MDP environment. If $n(s, a) \geq \frac{72S^3H^4R_{\max}^2 \ln(4/\delta')}{\epsilon^2}$, then with probability at least $1 - 2\delta'$, $|\hat{r}(s, a) - r(s, a)| \leq \epsilon_R$ and $||\hat{p}(s, a) - p(s, a)||_1 \leq \epsilon_T$. In particular, we consider $\epsilon_R = \frac{\epsilon}{6SH}$ and $\epsilon_T = \frac{\epsilon}{6SH^2R_{\max}}$.*

*Proof.*

For any state-action pair $(s, a)$, with probability $1 - 2\delta'$, Model-based Interval Estimation [64, 65] guarantees,

$$|\hat{r}(s, a) - r(s, a)| \leq \sqrt{\frac{\ln(4/\delta')R_{\max}^2}{2n(s, a)}} \tag{1.1}$$

and

$$||\hat{p}(s, a) - p(s, a)||_1 \leq \sqrt{\frac{2[\ln(2^S - 2) - \ln(\delta')]}{n(s, a)}}. \tag{1.2}$$

We wish to find the number of episodes, $m$, such that if $n(s, a) \geq m$, then $|\hat{r}(s, a) - r(s, a)| \leq \epsilon_R$ (1.1) and $||\hat{p}(s, a) - p(s, a)||_1 \leq \epsilon_T$ (1.2) are satisfied. We can therefore let

$$m \geq \max\left\{\frac{\ln(4/\delta')R_{\max}^2}{2(\epsilon_R)^2}, \frac{2[\ln(2^S - 2) + \ln(1/\delta')]}{(\epsilon_T)^2}\right\}. \tag{1.3}$$

Since $\ln(2^S - 2) \leq S$ and $a + b \leq ab$ if $a, b \geq 2$, we can let

$$m \geq \max\left\{\frac{\ln(4/\delta')R_{\max}^2}{2(\epsilon_R)^2}, \frac{2S\ln(1/\delta')}{(\epsilon_T)^2}\right\}. \tag{1.4}$$

As required by Lemma 4 (4.17), we choose $\epsilon_R \leq \frac{\epsilon}{6SH}$ and $\epsilon_T \leq \frac{\epsilon}{6SH^2R_{\max}}$, where $\epsilon$ corresponds to the suboptimality bound described in Theorem 2. Then

$$m \geq \max\left\{\frac{\ln(4/\delta')R_{\max}^2 \cdot 36S^2H^2}{2\epsilon^2}, \frac{2S\ln(1/\delta') \cdot 36S^2H^4R_{\max}^2}{\epsilon^2}\right\} \tag{1.5}$$

$$\implies m \geq \max\left\{\frac{18S^2H^2R_{\max}^2 \ln(4/\delta')}{\epsilon^2}, \frac{72S^3H^4R_{\max}^2 \ln(1/\delta')}{\epsilon^2}\right\} \tag{1.6}$$

$$\implies m \geq \frac{72S^3H^4R_{\max}^2 \ln(4/\delta')}{\epsilon^2}. \tag{1.7}$$

Equation (1.7) implies that if $n(s,a) \geq \frac{72 S^3 H^4 R_{\max}^2 \ln(4/\delta')}{\epsilon^2}$, then any $(s,a)$ has both bounded transition and reward model estimate errors, with probability at least $1 - 2\delta'$. $\square$

Next we want to determine a sufficient number of episodes $N_{(s,a)}$ such that if we run EULER for $N_{(s,a)}$ episodes on any state-action pair, $n(s,a) \geq \frac{72 S^3 H^4 R_{\max}^2 \ln(4/\delta')}{\epsilon^2}$ is guaranteed for any $(s,a)$ if that pair is $\zeta$-significant. This step closely follows the regret analysis of EULER algorithm [76] in the context of reward-free exploration used by Jin et al. [28].

**Lemma 2.** *With probability at least $1 - \delta'$, if EULER is executed for $N_0 = O(S^2 A H^{\frac{5}{2}} \iota^3 / \zeta)$ episodes, where $\iota := \log(SAH/\delta'\zeta)$, on any $\zeta$-significant state $s \in \mathcal{S}$ as the exploration target, then EULER finds a set of policies $\Phi^{(s)}$ with $|\Phi^{(s)}| = N_0$ such that $\frac{1}{2} \max_\pi P^\pi(s) \leq \frac{1}{N_0} \sum_{\pi \in \Phi^{(s,a)}} P^\pi(s)$.*

*Proof.*

Similar to the reward-free exploration analysis in Jin et al. [28], our proposed *Observe-then-Plan* also uses EULER as a black-box exploration algorithm to reach all $\zeta$-significant states and actions with high probability (Algorithm 3, L3:9). We show that with a sufficiently large $N_0$, the cumulative probabilities are guaranteed to satisfy $\frac{1}{2} \max_\pi P^\pi(s) \leq \frac{1}{N_0} \sum_{\pi \in \Phi^{(s,a)}} P^\pi(s)$.

For completeness, we begin with Proposition 6, Problem Independent Bound for EULER with Bernstein Inequality by Zanette and Brunskill [76], which shows with probability at least $1 - \delta'$, the regret of EULER at timestep T is bounded by

$$\widetilde{O}\left(\sqrt{\frac{G^2}{H} SAT} + \sqrt{S}SAH^2(\sqrt{S} + \sqrt{H})\right) \tag{2.1}$$

Lemma 3.4 by Jin et al. [28] shows that the upper bound $G^2$ in EULER can be replaced by $4V^*(s)$ because the reward function under the exploration objective is zero for all states except the target state $s$ and thus $\sum_{t=0}^{H-1} r(s_t) \leq 1$.

Using $V^*(s) = \max_\pi P^\pi(s)$ and $T = N_0 H$, we expand the regret bound (2.1) as,

$$\max_\pi P^\pi(s) - \frac{1}{N_0} \sum_{\pi \in \Phi^{(s)}} P^\pi(s) \leq c_0 \cdot \left\{ \sqrt{\frac{SA\iota \cdot \max_\pi P^\pi(s)}{N_0}} + \frac{S^2 A H^{\frac{5}{2}} \iota^3}{N_0} \right\} \tag{2.2}$$

for some constant $c_0$.

We want to find $N_0$ such that the right side of the inequality is bounded above by $\frac{1}{2} \max_\pi P^\pi(s)$, i.e.,

$$c_0 \cdot \left\{ \sqrt{\frac{SA\iota \cdot \max_\pi P^\pi(s)}{N_0}} + \frac{S^2 A H^{\frac{5}{2}} \iota^3}{N_0} \right\} \leq \frac{1}{2} \max_\pi P^\pi(s). \tag{2.3}$$

Following the proof of Theorem 3.3 by Jin et al. [28], we choose $N_0$ such that for some constant $0 < c_1 < 1$,

$$\sqrt{\frac{SA\iota \cdot \max_\pi P^\pi(s)}{N_0}} \leq c_1 \cdot \max_\pi P^\pi(s) \tag{2.4}$$

$$\frac{S^2 A H^{\frac{5}{2}} \iota^3}{N_0} \leq c_1 \cdot \max_\pi P^\pi(s). \tag{2.5}$$

Below we show that choosing $N_0 \geq \frac{S^2 A H^{\frac{5}{2}} \iota^3}{c_1^2 \max_\pi P^\pi(s)}$ is sufficient.

$$\sqrt{\frac{SA\iota \max_\pi P^\pi}{\frac{S^2 A H^{\frac{5}{2}} \iota^3}{c_1^2 \max_\pi P^\pi(s)}}} = c_1 \cdot \sqrt{\frac{1}{SH^{\frac{5}{2}} \iota^2}} \cdot \max_\pi P^\pi(s) \leq c_1 \cdot \max_\pi P^\pi(s). \tag{2.6}$$

$$\frac{S^2 A H^{\frac{5}{2}} \iota^3}{\frac{S^2 A H^{\frac{5}{2}} \iota^3}{c_1^2 \max_\pi P^\pi(s)}} = c_1^2 \cdot \max_\pi P^\pi(s) \leq c_1 \cdot \max_\pi P^\pi(s). \tag{2.7}$$

Thus if we choose $N_0 \geq \frac{S^2 A H^{\frac{5}{2}} \iota^3}{c_1^2 \max_\pi P^\pi(s)}$ for a sufficiently small constant $c_1$, we obtain $\max_\pi P^\pi(s) - \frac{1}{N_0} \sum_{\pi \in \Phi^{(s)}} P^\pi(s) \leq \frac{1}{2} \max_\pi P^\pi(s)$ and subsequently, $\frac{1}{2} \max_\pi P^\pi(s) \leq \frac{1}{N_0} \sum_{\pi \in \Phi^{(s)}} P^\pi(s)$.

Since $\max_\pi P^\pi(s) \geq \zeta$ for any $\zeta$-significant state by definition, if EULER is executed for $N_0 = O\left(\frac{S^2 A H^{\frac{5}{2}} \iota^3}{\zeta}\right)$ episodes, then $\frac{1}{N_0} \sum_{\pi \in \Phi^{(s)}} P^\pi(s)$, the expected probability of visiting $s$ while executing $\pi \in \Phi^{(s)}$ from the EULER-returned set of policies, is still at least as large as $\frac{1}{2} \max_\pi P^\pi(s) \geq \frac{\zeta}{2}$. Thus we have

$$\frac{1}{N_0} \sum_{\pi \in \Phi^{(s)}} P^\pi(s) \geq \frac{1}{2} \max_\pi P^\pi(s) \tag{2.8}$$

for all $\zeta$-significant states after $SAN_0$ episodes, where $N_0 = O\left(\frac{S^2 A H^{\frac{5}{2}} \iota^3}{\zeta}\right)$. $\square$

We conclude the exploration phase of *Observe-then-Plan* with Lemma 3 by running EULER on each target state-action pair for sufficiently many episodes to satisfy Lemma 1 and Lemma 2.

**Lemma 3.** *After $N_{(s,a)} = O\left(\max\left\{4\frac{72 S^3 H^4 R_{max}^2 \ln(4/\delta')}{\epsilon^2 \zeta}, \frac{S^2 A H^{\frac{5}{2}} \iota^3}{\zeta}, \frac{16 \ln(1/\delta')}{\zeta}\right\}\right)$ episodes of EU-LER, any $\zeta$-significant state-action $(s,a)$ is observed at least $\frac{72 S^3 H^4 R_{max}^2 \ln(4/\delta')}{\epsilon^2}$ times with probability at least $1 - \delta'$.*

*Proof.*

For any $\zeta$-significant state-action pair $(s,a)$, let $X_i$ be the indicator variable $\mathbf{1}[(s,a)$ is visited while following $\pi_i]$. Let $\sigma_i = \Pr\{X_i = 1\}$, and let $S_{N_{(s,a)}} = \sum_{i=1}^{N_{(s,a)}} X_i$, and $\mu = \sum_{i=1}^{N_{(s,a)}} \sigma_i$. We want to find $N_{(s,a)}$ such that with probability at least $1 - \delta'$, any $\zeta$-significant $(s,a)$ pair is observed at least $\frac{72 S^3 H^4 R_{max}^2 \ln(4/\delta')}{\epsilon^2}$ times after executing EULER for $N_{(s,a)}$ episodes.

First assume $N_{(s,a)} = O\left(\frac{S^2 A H^{\frac{5}{2}} \iota^3}{\zeta}\right)$ which provides the guarantee that $\mu = \sum_{i=1}^{N_{(s,a)}} \sigma_i \geq N_{(s,a)}(\frac{\zeta}{2})$ from Lemma 2.

Using the lower tail Chernoff bound on $S_{N_{(s,a)}}$ and $\mu$,

$$P\left(S_{N_{(s,a)}} \leq \frac{1}{2}\mu\right) \leq e^{(-\mu/8)} \tag{3.1}$$

Replace $\mu = N_{(s,a)}(\frac{\zeta}{2})$ and set the failure probability on the right side to be bounded by $\delta'$.

$$e^{(-N_{(s,a)} \frac{\zeta}{16})} \leq \delta' \tag{3.2}$$

$$N_{(s,a)} \frac{\zeta}{16} \geq \ln(1/\delta') \tag{3.3}$$

$$N_{(s,a)} \geq \frac{16 \ln(1/\delta')}{\zeta} \tag{3.4}$$

We also want to satisfy

$$\frac{\mu}{2} \geq N_{(s,a)}(\frac{\zeta}{4}) \geq \frac{72 S^3 H^4 R_{max}^2 \ln(4/\delta')}{\epsilon^2} \tag{3.5}$$

to ensure that any $\zeta$-significant $(s,a)$ pair is visited at least $\frac{72 S^3 H^4 R_{max}^2 \ln(4/\delta')}{\epsilon^2}$ times. Equation (3.5) guarantees that the error probability from Equation (3.1) is an upper bound on $Pr\{S_{N_{(s,a)}} < \frac{72 S^3 H^4 R_{max}^2 \ln(4/\delta')}{\epsilon^2}\}$ since $\{S_{N_{(s,a)}} < \frac{72 S^3 H^4 R_{max}^2 \ln(4/\delta')}{\epsilon^2}\} \subset \{S_{N_{(s,a)}} < \frac{\mu}{2}\}$.

Determine $N_{(s,a)}$ as,

$$N_{(s,a)} \geq \max \left\{ 4 \frac{72 S^3 H^4 R_{\max}^2 \ln (4/\delta')}{\epsilon^2 \zeta}, \frac{16 \ln (1/\delta')}{\zeta}, \frac{S^2 A H^{\frac{5}{2}} \iota^3}{\zeta} \right\} \tag{3.6}$$

After $N_{(s,a)} \geq \frac{288 S^3 A H^4 R_{\max}^2 \iota^3 \ln (4/\delta')}{\epsilon^2 \zeta}$ episodes, EULER observes at least $\frac{72 S^3 H^4 R_{\max}^2 \ln (4/\delta')}{\epsilon^2}$ samples of any $\zeta$-significant $(s,a)$ with probability at least $1 - \delta'$. We choose $\delta' = \frac{\delta}{4SA}$ and $\zeta = \frac{\epsilon}{18 S H R_{\max}}$, as will be discussed in Lemma 4, so $N_{(s,a)} \geq 288 \cdot 18 \cdot \frac{S^4 A H^5 R_{\max}^3 \iota^3 \ln (16SA/\delta)}{\epsilon^2}$. We multiply $N_{(s,a)}$, the necessary number of episodes for running EULER on a single $(s,a)$ pair as the exploration target, by $SA$ in order to guarantee that all $\zeta$-significant states and actions are visited at least $\frac{72 S^3 H^4 R_{\max}^2 \ln (4/\delta')}{\epsilon^2}$ times. $\square$

After the exploration phase of *Observe-then-Plan*, we use the transition and reward model estimates to plan a $t$-step conditional policy $\pi_t$ such that for any known initial state distribution, the $\alpha$-vector difference is bounded in terms of the model estimate errors.

**Lemma 4.** *Suppose we have approximate ACNO-MDP model parameters with errors $\|\hat{p}(s,a) - p(s,a)\|_1 \leq \epsilon_T$ and $|\hat{r}(s,a) - r(s,a)| \leq \epsilon_R$ for all $\zeta$-significant states and actions, then for any $t$-step conditional policy $\pi_t$ starting from any initial state $s_0$, $|\hat{\alpha}_t^{\pi_t}(s_0) - \alpha_t^{\pi_t}(s_0)| \leq \epsilon_R S t + \epsilon_T S R_{\max} t^2 + 3 \zeta S R_{\max} t$.*

*Proof.*

Separate every state $s \in S$ into $\zeta$-significant states (call this set $K$) and $\zeta$-insignificant states (call this set $\tilde{K}$). This proof closely follows the analysis by Jin et al. [28], leveraging the separation of $\zeta$-significant versus $\zeta$-insignificant states. Use the guarantee by Lemma 3 that all $\zeta$-significant states and actions have error-bounded model estimates, and for any $\zeta$-insignificant states and actions, the $\alpha$-vector estimates of those states have negligible effect on the final value function estimate.

First, we provide a version of the Value Difference Lemma by Dann et al. [11] (Lemma E.15) (also stated by Jin et al. [28] (Lemma C.1)). Unlike these previous expositions, we write the Value Difference Lemma with $\alpha$-vectors instead of with $V$ estimates. From Definition 3 (eq 2), we note the $\alpha$-vector definition in ACNO-MDP settings does not require the summation over every observation in $O$. Consider any $t$-step conditional policy $\pi_t$ and its $\alpha$-vector estimates, $\hat{\alpha}_t$, from any initial state $s_0$. If $t = 0$, we know $|\hat{\alpha}_t^{\pi_t}(s_0) - \alpha_t^{\pi_t}(s_0)| = |\hat{\alpha}_0^{\pi_0}(s_0) - \alpha_0^{\pi_0}(s_0)| = 0$ since there is no actionable step.

For any $t \geq 1$ and ACNO-MDP $\mathcal{M}$, the Value Difference Lemma states that

$$|\hat{\alpha}_t^{\pi_t}(s_0) - \alpha_t^{\pi_t}(s_0)| \tag{4.1}$$

$$= \mathbb{E}_{\mathcal{M}} \left[ \left\{ \left| \sum_{i=1}^t (\hat{r}(s_i, a_i) - r(s_i, a_i) + \sum_{s'} (\hat{p}(s'|s_i, a_i) - p(s'|s_i, a_i)) \cdot \hat{\alpha}_{i-1}(s')) \right| \right\} | s_0 \right] \tag{4.2}$$

$$\leq \mathbb{E}_{\mathcal{M}} \left[ \left\{ \sum_{i=1}^t (|\hat{r}(s_i, a_i) - r(s_i, a_i)| + \sum_{s'} |(\hat{p}(s'|s_i, a_i) - p(s'|s_i, a_i)| \cdot \hat{\alpha}_{i-1}(s')) \right\} | s_0 \right] \tag{4.3}$$

since $\hat{\alpha}_{i-1}(s) \geq 0$ for any $s$ given $r(s,a) \in [0, R_{\max}]$, as stated in Theorem 2.

By definition of expectation, we can express the inequality 4.2 as follows:

$$|\hat{\alpha}_t^{\pi_t}(s_0) - \alpha_t^{\pi_t}(s_0)| \tag{4.4}$$

$$\leq \sum_{i=1}^t \sum_{s_i \in S} P(s_i|s_0; \pi) (|\hat{r}(s_i, a_i) - r(s_i, a_i)| + \sum_{s'} |(\hat{p}(s'|s_i, a_i) - p(s'|s_i, a_i)| \cdot \hat{\alpha}_{i-1}(s')) \tag{4.5}$$

First consider the reward difference term.

$$\sum_{i=1}^t \sum_{s_i \in S} P(s_i|s_0; \pi) |\hat{r}(s_i, a_i) - r(s_i, a_i)| \tag{4.6}$$

Separate every $s_i \in S$ into either $K$ or $\tilde{K}$:

$$\sum_{i=1}^{t} \sum_{s_i \in S} P(s_i|s_0; \pi)|\hat{r}(s_i, a_i) - r(s_i, a_i)| \tag{4.7}$$

$$= \sum_{i=1}^{t} \sum_{s_i \in K} P(s_i|s_0; \pi)|\hat{r}(s_i, a_i) - r(s_i, a_i)| + \sum_{s_i \in \widetilde{K}} P(s_i|s_0; \pi)|\hat{r}(s_i, a_i) - r(s_i, a_i)| \tag{4.8}$$

Since the reward estimate error is bounded above by $\epsilon_R$ for every $\zeta$-significant state-action pair and $|\hat{r}(s_i, a_i) - r(s_i, a_i)| \leq |\hat{r}(s_i, a_i)| \leq R_{\max}$, we obtain

$$\sum_{i=1}^{t} \sum_{s_i \in S} P(s_i|s_0; \pi)|\hat{r}(s_i, a_i) - r(s_i, a_i)| \leq \sum_{i=1}^{t} \sum_{s_i \in K} P(s_i|s_0; \pi)\epsilon_R + \sum_{i=1}^{t} \sum_{s_i \in \widetilde{K}} P(s_i|s_0; \pi)R_{\max}. \tag{4.9}$$

Because $|K| \leq S$ and the order of summation can be interchanged,

$$\sum_{i=1}^{t} \sum_{s_i \in K} P(s_i|s_0; \pi)\epsilon_R + \sum_{i=1}^{t} \sum_{s_i \in \widetilde{K}} P(s_i|s_0; \pi)R_{\max} \leq tS\epsilon_R + R_{\max} \sum_{s_i \in \widetilde{K}} \sum_{i=1}^{t} P(s_i|s_0; \pi) \tag{4.10}$$

By definition of $\zeta$-insignificant state, $\sum_{i=1}^{t} P^{\pi}(s_i|s_0) < \zeta$ for any policy $\pi$ if $s_i \in \widetilde{K}$ and $|\widetilde{K}| \leq S$. Thus

$$tS\epsilon_R + R_{\max} \sum_{s_i \in \tilde{K}} \sum_{i=1}^{t} P(s_i|s_0; \pi) \leq tS\epsilon_R + S\zeta R_{\max} \tag{4.11}$$

Next consider the transition difference term:

$$\sum_{i=1}^{t} \sum_{s_i \in S} P(s_i|s_0; \pi) \left\{ \sum_{s'} |(\hat{p}(s'|s_i, a_i) - p(s'|s_i, a_i)| \cdot \hat{\alpha}_{i-1}(s') \right\}. \tag{4.12}$$

Similarly separate all $s_i \in S$ into either $K$ or $\tilde{K}$.

$$\sum_{i=1}^{t} \left\{ \sum_{s_i \in K} P(s_i|s_0; \pi) \sum_{s'} |(\hat{p}(s'|s_i, a_i) - p(s'|s_i, a_i)| \cdot \hat{\alpha}_{i-1}(s') \right\} + \tag{4.13}$$

$$\sum_{i=1}^{t} \left\{ \sum_{s_i \in \widetilde{K}} P(s_i|s_0; \pi) \sum_{s'} |(\hat{p}(s'|s_i, a_i) - p(s'|s_i, a_i)| \cdot \hat{\alpha}_{i-1}(s') \right\} \tag{4.14}$$

$$\leq \sum_{i=1}^{t} \left\{ \sum_{s_i \in K} \epsilon_T \cdot \max_{s'} \hat{\alpha}_{i-1}(s') + \sum_{s_i \in \widetilde{K}} P(s_i|s_0; \pi) \sum_{s'} |(\hat{p}(s'|s_i, a_i) - p(s'|s_i, a_i)| \cdot \hat{\alpha}_{i-1}(s') \right\} \tag{4.15}$$

$$\leq \sum_{i=1}^{t} \left\{ \sum_{s_i \in K} \epsilon_T \cdot R_{\max}i + \sum_{s_i \in \tilde{K}} P(s_i|s_0; \pi) \cdot 2 \cdot R_{\max}i \right\} \tag{4.16}$$

$$= \left\{ \sum_{i=1}^{t} \sum_{s_i \in K} \epsilon_T \cdot R_{\max}i \right\} + \left\{ \sum_{s_i \in \tilde{K}} \left( \sum_{i=1}^{t} P(s_i|s_0; \pi) \right) \cdot 2 \cdot R_{\max}t \right\} \tag{4.17}$$

where line 4.12 follows from the fact that the transition estimate error is bounded above by $\epsilon_T$ for all $\zeta$-significant state-action pairs and line 4.13 follows from the facts that $\max \hat{\alpha}_{i-1} \leq R_{\max}i$ and $\sum_{s'} |\hat{p}(s'|s_i, a_i) - p(s'|s_i, a_i)| \leq \sum_{s'} |\hat{p}(s'|s_i, a_i)| + |p(s'|s_i, a_i)| = ||\hat{p}(s_i, a_i)||_1 + ||p(s_i, a_i)||_1 = 2$.

By the definition of $\zeta$-insignificant states,

$$\left\{ \sum_{i=1}^{t} \sum_{s_i \in K} \epsilon_T \cdot R_{\max}i \right\} + \left\{ \sum_{s_i \in \tilde{K}} \left( \sum_{i=1}^{t} P(s_i|s_0; \pi) \right) \cdot 2 \cdot R_{\max}t \right\} \tag{4.18}$$

$$\leq \left\{ \sum_{i=1}^{t} \sum_{s_i \in K} \epsilon_T \cdot R_{\max} i \right\} + \left\{ \sum_{s_i \in \tilde{K}} \zeta \cdot 2 \cdot R_{\max} t \right\}. \tag{4.19}$$

$$\leq t S \epsilon_T R_{\max} t + 2 S \zeta R_{\max} t \tag{4.20}$$

Combining the bounds from (4.8) and (4.16), we conclude

$$|\hat{\alpha}_t^{\pi_t}(s_0) - \alpha_t^{\pi_t}(s_0)| \leq \epsilon_R S t + S \zeta R_{\max} + \epsilon_T S R_{\max} t^2 + 2 \zeta S R_{\max} t \tag{4.21}$$

$$\leq \epsilon_R S t + \epsilon_T S R_{\max} t^2 + 3 \zeta S R_{\max} t. \tag{4.22}$$

By choosing $\zeta = \frac{\epsilon}{18 S H R_{\max}}, \epsilon_R \leq \frac{\epsilon}{6SH}, \epsilon_T \leq \frac{\epsilon}{6SH^2 R_{\max}}$, we have for any $t \leq H$,

$$|\hat{\alpha}_t^{\pi_t}(s_0) - \alpha_t^{\pi_t}(s_0)| \leq \left( \frac{\epsilon}{6SH} \right) SH + \left( \frac{\epsilon}{6SH^2 R_{\max}} \right) SH^2 R_{\max} + \left( \frac{\epsilon}{18 S H R_{\max}} \right) 3 SH R_{\max} = \frac{\epsilon}{2} \tag{4.23}$$

$\square$

Once the $\alpha$-vector estimates are bounded, we closely follow the proof of EEPORL (217-229) by Guo et al. [21] to show the value function estimates are near-accurate as shown below.

**Lemma 5.** *If an $\alpha$-vector for state $s_0$ following any $t$-step conditional ACNO-MDP policy $\pi_t$ has bounded error $|\hat{\alpha}_t^{\pi_t}(s_0) - \alpha_t^{\pi_t}(s_0)| \leq \frac{\epsilon}{2}$ and the initial belief $b_0$ (i.e., state distribution) is known, then $V^*(b_0) - V^{\hat{\pi}}(b_0) \leq \epsilon$ for the true optimal policy $\pi^*$ and returned policy $\hat{\pi}$.*

*Proof.*

$$V^*(b_0) - V^{\hat{\pi}}(b_0) = b_0 \cdot \alpha^{\pi^*} - b_0 \cdot \alpha^{\hat{\pi}} \tag{4.1}$$

$$= b_0 \cdot \alpha^{\pi^*} - b_0 \cdot \alpha^{\hat{\pi}} + (b_0 \cdot \hat{\alpha}^{\hat{\pi}} - b_0 \cdot \hat{\alpha}^{\hat{\pi}}) + (b_0 \cdot \hat{\alpha}^{\pi^*} - b_0 \cdot \hat{\alpha}^{\pi^*}) \tag{4.2}$$

$$= b_0 \cdot (-\alpha^{\hat{\pi}} + \hat{\alpha}^{\hat{\pi}}) + b_0 \cdot (\hat{\alpha}^{\pi^*} - \hat{\alpha}^{\hat{\pi}}) + b_0 \cdot (\alpha^{\pi^*} - \hat{\alpha}^{\pi^*}) \tag{4.3}$$

Since $\hat{\pi}$ is found optimal under $\hat{\alpha}$-vectors, $\hat{\alpha}^{\hat{\pi}} \geq \hat{\alpha}^{\pi^*}$ and therefore

$$b_0 \cdot (-\alpha^{\hat{\pi}} + \hat{\alpha}^{\hat{\pi}}) + b_0 \cdot (\hat{\alpha}^{\pi^*} - \hat{\alpha}^{\hat{\pi}}) + b_0 \cdot (\alpha^{\pi^*} - \hat{\alpha}^{\pi^*}) \leq b_0 \cdot (-\alpha^{\hat{\pi}} + \hat{\alpha}^{\hat{\pi}}) + b_0 \cdot (\hat{\alpha}^{\hat{\pi}} - \hat{\alpha}^{\hat{\pi}}) + b_0 \cdot (\alpha^{\pi^*} - \hat{\alpha}^{\pi^*}) \tag{4.4}$$

$$\leq b_0 \cdot |\hat{\alpha}^{\hat{\pi}} - \alpha^{\hat{\pi}}| + b_0 \cdot |\hat{\alpha}^{\pi^*} - \alpha^{\pi^*}|. \tag{4.5}$$

Since $b_0 \in \mathbb{R}^{|\mathcal{S}|}$ is an initial probability distribution over the states and $\alpha \in \mathbb{R}^{|\mathcal{S}|}$ is also defined over the states,

$$b_0 \cdot |\hat{\alpha}^{\hat{\pi}} - \alpha^{\hat{\pi}}| + b_0 \cdot |\hat{\alpha}^{\pi^*} - \alpha^{\pi^*}| \leq ||\hat{\alpha}^{\hat{\pi}} - \alpha^{\hat{\pi}}||_\infty + ||\hat{\alpha}^{\pi^*} - \alpha^{\pi^*}||_\infty. \tag{4.6}$$

Thus

$$V^*(b_0) - V^{\hat{\pi}}(b_0) \leq ||\hat{\alpha}^{\hat{\pi}} - \alpha^{\hat{\pi}}||_\infty + ||\hat{\alpha}^{\pi^*} - \alpha^{\pi^*}||_\infty. \tag{4.7}$$

For any policy $\pi_t$ and initial state $s_0 \in \mathcal{S}$, $|\hat{\alpha}_t^{\pi_t}(s_0) - \alpha_t^{\pi_t}(s_0)| \leq \frac{\epsilon}{2}$ by Lemma 4, so the $\ell_\infty$-norm over the $\alpha$-vectors is also bounded above by $\frac{\epsilon}{2}$. Therefore $V^*(b_0) - V^{\hat{\pi}}(b_0)$ is bounded above by $\epsilon$. This guarantees that after $O\left( \frac{S^5 A^2 H^5 R_{\max}^3 t^4}{\epsilon^3} \right)$ episodes, with probability at least $1 - \delta$, the bounded $\alpha$-vector estimates yield near-accurate value function estimates. By finding an optimal policy under these value function estimates, *Observe-then-Plan* guarantees that the returned $t$-step ACNO-MDP policy is $\epsilon$-optimal with high probability. $\square$

# Appendix B. Experimental details

Our code is available on https://github.com/nam630/acno_mdp.

## Appendix B.1 Algorithms for tabular states and actions

We describe the training details of the algorithms used to obtain a policy for treating a Sepsis patient. At the end of training episodes, the estimated transition and reward model (or $Q$-function estimates), are used to evaluate the final policy under 50 rollouts. (Evaluation results are included in Table 3 of the manuscript.)

**Partially observable Monte-carlo tree search (POMCP) details**   *Observe-then-Plan* and ACNO-POMCP plan optimal action trajectories using POMCP. In particular, ACNO-POMCP uses POMCP to simulate rollouts under the transition and reward models based on samples observed during the exploration phase of *Observe-then-Plan*. As the *Observe-then-Plan* algorithm decouples exploration from planning, these estimated transition and reward model parameters are no longer updated once the algorithm enters the planning phase following an initial 1,000 episodes of exploration.

We use the POMDPy implementation of POMCP [14] with the following parameters: 2000 particles at the root node, 5000 roll-outs, and a max search depth of 5. During simulation, actions are selected by UCB1, which adds an uncertainty bonus to $Q$ values as $Q^{\oplus}(h, a) = Q(h, a) + c\sqrt{\frac{\log N(h)}{N(h,a)}}$ with coefficient $c = 3$. The action resulting in the highest estimated reward (using information aggregated from simulated rollouts) is the action that the agent executes in the true environment.

The best next action is recomputed after each step in the environment. In summary, our process follows: First start with a set of 2,000 initial belief states, simulate trajectories by choosing actions based on UCB from the initial state, take an action in the real environment and collect observations (either missing or of the true underlying states) and rewards, re-sample previous belief particles to keep them consistent with the received observation, simulate trajectories again from the resampled belief particle set. If the agent chooses to not observe, every belief particle is kept between steps in the environment because the missing observation cannot completely rule out any states from its belief set as implausible; on the other hand, if the agent chooses to observe, resampling rejects any particles whose next state observation would not match the true state observation. In order to speed up the runtime, we additionally set a particle selection timeout of 0.2 second per step if the particle set has at least 3 belief particles.

**EULER exploration details for *Observe-then-Plan***   We implement Algorithm 2 of EULER [76] with the following adjustments to the originally proposed parameters: $\ln \frac{4SAT}{\delta'}$ (denoted by $C$ hereafter)$= 0.01, H = 5, B_v = \sqrt{2C}, J = H * C/3, B_p = HB_v$. Note $S = 720, A = 8$ in Sepsis but for practical reasons, we approximate the log failure probability to be 0.01 and adjust the relevant parameters (e.g., $J, B_v, B_p$) accordingly. Additionally, we modify the reward function following the reward-free exploration approach proposed by Jin et al. [28], but we assume transition dynamics are stationary. First, fix an arbitrary state-action pair $(s, a)$. Construct an auxiliary MDP $\mathcal{M}'$ where the reward is 1 only for the chosen $(s, a)$ pair and 0 for all other states and actions. Also make $s$ a terminal state in $\mathcal{M}'$. While executing EULER, every actionable step is observed and the tuple $(s, a, s', r)$ is collected into the dataset. While our theoretical guarantees would require a much larger number of episodes, empirically on the Sepsis simulator, we observe that after 1k episodes, the transition and reward model estimates are sufficiently good for learning an optimal policy using POMCP.

**EULER exploration details for EULER-VI**   We use the same implementation for EULER as *Observe-then-Plan*. The main difference is that actions in EULER-VI are chosen based on the confidence intervals of the value estimates calculated from the observed rewards from the true environment $\mathcal{M}$ instead of the modified reward function used only during reward-free exploration [28] under $\mathcal{M}'$. Once the transition and reward model estimates are learned after 1k episodes, standard value iteration is computed under the learned model to extract the highest reward policy, which maps every state at a given timestep to some action. During evaluation, same as with other algorithms, we evaluate the value-iteration policy with 50 rollouts in the real environment but additionally assuming the observation cost is incurred at every actionable timestep since the standard value iteration assumes the environment is an MDP.

**ACNO-POMCP training details**   The transition model is initialized to uniform distributions over the entire state space, and the reward function is initialized to 0. The algorithm maintains $n(s, a), n(s, a, s'), r(s, a)$ and estimates the transition $p(s, a, s')$ as $\frac{n(s,a,s')}{n(s,a)}$ and the average instantaneous reward as $\frac{\sum_i r_i(s,a)}{n(s,a)}$. Every time a tuple $(s, a, s', r)$ is observed, the associated counts are incremented and the model is recomputed at the end of every episode if there is any changes to the counts. Unlike *Observe-then-Plan* which only chooses greedy actions to execute in the real environment, ACNO-POMCP selects actions by $\epsilon$-greedy to explore the real environment. In particular, ACNO-POMCP uses an initial $\epsilon$ value of 1.0 and minimum of 0.1 with a decaying factor of 0.95

applied after every episode. ACNO-POMCP does not require any fixed number of observations to be made prior to planning since whether and when to observe the outcome of actions is determined by the online agent. The total training episode length is 2000, same with DRQN.

**DRQN training details**   DRQN takes in as input a length 721 one-hot encoded vector of the patient state (720 states for patient state and additional 1 for a missingness observation). The network has one hidden layer of size 16, one LSTM layer of 16 hidden weights. The LSTM output is passed through another hidden layer of size [16] followed by ReLU activation and mapped to the 16 values each representing a discrete action choice. The DRQN network parameters are optimized with Adam using a learning rate of 1e-3. The buffer size is kept at the maximum number of training episodes. For exploration, we use $\epsilon$-greedy with the initial value of 1 and a decaying factor of 0.999 applied after every episode. The Q estimate parameters are updated at the end of every episode for a total of 2k training episodes. Only greedy actions are selected with $\epsilon = 0$ during the 50 evaluation episodes.

**Appendix B.2 Algorithms for continuous states and actions**

**A2C training details**   Same as the original implementation for DVRL by Maximilian et al. [43], our implementation uses 16 parallel environments and 5 steps for update. In total, we train for $2.5 \times 10^7$ frames (steps). The encoder, the policy, and the value function are trained end-to-end with the following weights to the loss function: action coefficient $\lambda^A = 1$, entropy coefficient $\lambda^H = 0.01$, value loss coefficient $\lambda^V = 0.5$, and encoding loss coefficient $\lambda^E = 1$, and the loss function at timestep $t$ is defined as $\mathcal{L}_t = \lambda^A \mathcal{L}_t^A + \lambda^H \mathcal{L}_t^H + \lambda^V \mathcal{L}_t^V + \lambda^E \mathcal{L}_t^E$ [43], which is a standard A2C loss function plus the belief encoding loss $\mathcal{L}^E$. For the standard A2C implementation (under the always observing assumption), we simply exclude $\mathcal{L}^E$ and use the state observations as inputs to the policy and value function networks. We use RMSProp optimizer with $\alpha = 0.99$, max gradient norm = 0.5, and a learning rate of $0.0001$.

The policy is one fully connected layer with 5 outputs (i.e., 2 for Guassian mean, 2 for standard deviation, and 1 binary output for whether to observe or not) on Mountain Hike and 4 (i.e., {Left, Right} × {Observe, Not observe}) on Cart Pole. The observing action part of the policy for Mountain Hike is passed through a Sigmoid layer and converted to 0 (if $\leq 0.5$) if not observing and 1 (if > 0.5) if observing. The output of the Cart Pole policy is passed through a Softmax layer to generate a probability distribution over the action space. The value function is also a fully connected layer whose input size is the same as the size of an RNN latent state $h_t$ and output size is always 1.

**Belief encoder update**   Both encoders update 30 belief particles $(= K)$ and each belief at timestep $t$ is represented by a latent state $z_t$, an RNN latent state $h_t$, and a weight $w_t$. For learning the actor and the critic, the belief particles are passed through one GRU layer and encoded into a single aggregated 1d array $\hat{h}_t$ which has the same size as $h_t$. The policy and value function networks are defined over the aggregated $\hat{h}_t$. Below we describe vanilla DVRL belief encoder [43] and our modified version of ACNO-belief encoder. While other approaches, such as Bayesian filtering, may be considered for state estimation, the original work on DVRL points out that many existing methods are not scalable to large or continuous states and observations. Their paper also includes ablation studies on the influence of belief particle resampling after weights are readjusted according to the received observation, and a comparison of DVRL encoders with RNN encoders which aggregate observations and actions into a belief history without any reconstruction loss (see Algorithms 1 and 2 of Appendix in the original DVRL paper [43]).

**DVRL belief encoder model architecture**   DVRL [43] has three encoders for observations $\varphi^o$, latent states $\varphi^z$ and actions $\varphi^a$, four networks for belief update (i.e., proposal, emission, transition, and deterministic transition) and one network for belief aggregation to output $\hat{h}_t$. The observation encoder $\varphi^o$ has two fully connected layers of size [64, 64], each followed by ReLU activation, and we use an observation encoding of size 64. The action encoder $\varphi^a$, which outputs an encoding of dimension 128, is one fully connected layer with ReLU activation. The latent state $z_t$ (of size 128 on Mountain Hike and 256 on Cart Pole) is also passed through one fully connected layer and ReLU activation to generate $\varphi^z$. The size of $z_t$ is the same as the size of an RNN latent state $h_t$ as well as the aggregated belief state $\hat{h}_t$. The proposal network $q_\theta(z_t|h_{t-1}, x^o, x^a)$ is one fully connected layer with ReLU activation with two heads that additionally takes in an encoded observation $x^o$ and an encoded action $x^a$. The latent state is assumed Gaussian, so one head outputs a mean and the other head

outputs a standard deviation. The emission network $p_\theta(o_t|h_t, x^z, x^a)$ is one fully connected layer with ReLU activation where $x^z$ is an encoded latent state. The transition network $p_\theta(z_t|h_{t-1}, x^a)$ has one hidden layer of size $[h_t]$ with ReLU activation and two heads simiarly for outputting a mean and a standard deviation. The last transition network is a GRU cell that outputs the aggregated $h_t$ from the concatenated encodings of latent states, observations, and actions, $(h_{t-1}, x^z, x^o, x^a)$.

This pseudocode is from DVRL by $Maximilian\, et\, al.$ [43] Algorithm 1 DVRL encoder.

---

**Algorithm 4** DVRL encoder

---

    **Input** Previous state belief $\hat{b}_{t-1}$, observation $o_t$, action $a_{t-1}$.
    **Output** $\hat{b}_t$, and model estimation loss $\mathcal{L}_t^{\text{ELBO}}$.
1: Unpack $w_{t-1}^{1:K}, z_{t-1}^{1:K}, h_{t-1}^{1:K}, \hat{h}_{t-1} \leftarrow \hat{b}_{t-1}$
2: $x^o \leftarrow \varphi_\theta^o(o_t)$
3: $x^a \leftarrow \varphi_\theta^a(a_{t-1})$
4: **for** every particle $k = 1, \dots, K$ **do**
5:     Resample $h_{t-1}^k \sim h_{t-1}^{1:K}$ based on weights
6:     Sample $z_t^k \sim q_\theta(z_t^k|h_{t-1}^k, x^o, x^a)$
7:     $x^z \leftarrow \varphi_\theta^z(z_t)$
8:     $w_t^k \leftarrow p_\theta(z_t^k|h_{t-1}^k, x^a)p_\theta(o_t|h_{t-1}^k, x^z, x^a)/q_\theta(z_t^k|h_{t-1}^k, x^o, x^a)$
9:     $h_t^k \leftarrow \text{GRU}(h_{t-1}^k, x^z, x^o, x^a)$
10: **end for**
11: $\mathcal{L}_t^{\text{ELBO}} \leftarrow -\log \sum_k w_t^k - \log(K)$
12: $\hat{h}_t \leftarrow \text{GRU}(\text{Concat}(w_t^k, z_t^k, h_t^k)_{k=1}^K \text{passed sequentially})$
13: Pack $\hat{b}_t \leftarrow (w_t^{1:K}, z_t^{1:K}, h_t^{1:K}, \hat{h}_t)$

---

**ACNO-A2C belief encoder model architecture**    Our model also has the same encoders $\varphi^o, \varphi^z, \varphi^a$ with the same architecture as the DVRL model. We also use the same one layer GRU for belief aggregation to output $\hat{h}_t$. However, for belief update, our encoder only has two networks: one for predicting the next state and the other for ouputting an aggregated latent state $h_t$. We additionally leverage that $\dim(z_t) = \dim(o_t)$ since $o_t$ gives the true underlying state $z_t$, but keep the same aggregate state $h_t$ dimensions as the original DVRL (e.g., $\dim(h_t) = 128$ on Mountain Hike and 256 on Cart Pole). The first network $p_\theta(z_t|h_{t-1}, x^a)$ has one hidden layer of size $[h_t]$ followed by ReLU activation, and two heads for outputting a mean and a standard deviation. The network for outputting $h_t$ has the same architecture as DVRL, and the GRU network aggregates every concatenated encoding into $\hat{h}_t$. We only include the belief encoding loss $\mathcal{L}_t^E$ to the total loss $\mathcal{L}_t$ if $o_t$ is observed, and no encoding loss is added if the underlying state is not observed.

This pseudocode modifies the original implementation of the DVRL encoder to incorporate the ACNO-MDP observation structure into the inference network.

**Algorithm 5** ACNO-A2C belief encoder (including encoding networks $\varphi^{a,o,z}$)

**Input** Previous state belief $\hat{b}_{t-1}$, observation $o_t$, action $a_{t-1}$.

**Output** $\hat{b}_t$, and next state prediction loss $\mathcal{L}_t$.

1: Unpack $w_{t-1}^{1:K}, z_{t-1}^{1:K}, h_{t-1}^{1:K}, \hat{h}_{t-1} \leftarrow \hat{b}_{t-1}$
2: $x^o \leftarrow \varphi_\theta^o(o_t)$
3: $x^a \leftarrow \varphi_\theta^a(a_{t-1})$
4: **for** every particle $k = 1, \ldots, K$ **do**
5:     Resample $h_{t-1}^k \sim h_{t-1}^{1:K}$ based on weights
6:     **if** $a_{t-1} =$ observe **then**
7:         $z_t^k = o_t, w_t^k = \frac{1}{K}$
8:         $\mathcal{L}_t \leftarrow \mathcal{L}_t - \log p_\theta(o_t | h_{t-1}^k, x^a)$
9:     **else**
10:         Sample $z_t^k \sim p_\theta(z_t^k | h_{t-1}^k, x^a)$
11:         $w_t^k \leftarrow p_\theta(z_t^k | h_{t-1}^k, x^a)$
12:     **end if**
13:     $x^z \leftarrow \varphi_\theta^z(z_t)$
14:     $h_t^k \leftarrow \text{GRU}(h_{t-1}^k, x^z, x^o, x^a)$
15: **end for**
16: $\hat{h}_t \leftarrow \text{GRU}(\text{Concat}(w_t^k, z_t^k, h_t^k)_{k=1}^K \text{passed sequentially})$
17: Pack $\hat{b}_t \leftarrow w_t^{1:K}, z_t^{1:K}, h_t^{1:K}, \hat{h}_t$ and include $\mathcal{L}_t$ if $a_{t-1} =$ observe.

**Standard A2C model architecture**   Standard A2C assumes that the observation is made at every timestep and uses a state encoder which maps the observation input into an encoding dimension. Here we still use the same dimensions for the hidden state as the above two algorithms (e.g., 128 on Mountain Hike and 256 on Cart Pole) followed by ReLU activation. The encoded output is passed through a single layer linear policy and a value function network. The policy network uses a Softmax layer to output a distribution over the two action choices on Cart Pole (e.g., $|A| = 2$ assuming every step is observed) and 2-dimensional Gaussian mean and also 2-dimensional standard deviation outputs for actions on Mountain Hike, and a fixed observation cost is applied to every actionable step in the environment.

**Appendix B.3 Experimental domains**

**Sepsis**   The patient starts from the initial state consisting of {heart rate = normal, blood pressure = low, oxygen concentration = low, glucose level = normal, antibiotic state = off, vasopressor state = off, ventilation state = off} which is represented by a tabular value "256". $S = 720$ and $A = 8$ (treatment choices)$*2$ (option of observing or not) $= 16$ for choosing both whether to observe or not and which treatment option to take. We assume the patient is non-diabetic and model the same stochastic transitions used in the Sepsis simulator code by Oberst and Sontag [48]. Every episode has maximum 5 actionable steps and may terminate early if the patient is discharged or shows more than 3 abnormal symptoms. As described in the main text, a reward of 1 is given for discharge, 0 for death, and 0.25 for neutral states. We use a discount factor of 0.7.

**Mountain Hike**   Mountain Hike is run for 75 steps per episode with a discount factor of 0.99. In the code implementation of DVRL by Maximilian et al. [43], the transition is defined as: $(x_{t+1}, y_{t+1}) = (x_t, y_t) + (\triangle x_t, \triangle y_t) + N(0, 0.025 \cdot I_{2 \times 2})$ where $(\triangle x_t, \triangle y_t)$ is given by action $a_t$. Goal position is $(0.7, 0.5)$ with radius $(0.1)$. The starting x, y coordinates are drawn independently from $N(-0.85, 0.1)$. We use 64 for the action encoding and 128 for the dimensions of both the latent state $h_t$ and the aggregated belief state $\hat{h}_t$.

**Cart Pole**   Cart Pole is run for maximum 200 steps per episode. An episode terminates early if the pole falls more than $\pm 12$ degrees angle or the cart position is more than $\pm 2.4$ from the center of the screen. A reward of 1 is given to every surviving step with a discount factor of 0.98. The action encoding has 128 dimensions and both the hidden state $h_t$ and aggregated hidden state $\hat{h}_t$ have 256 dimensions.

## Appendix B.4 Experiment runtime

We include the logged wall clock runtimes of running different algorithms on the simulated/toy environments below:

| Environment | Algorithm | Experiment time (hr:min:sec) |
|---|---|---|
| Sepsis | *Observe-then-Plan* | 9:52:0.66 |
| | ACNO-POMCP | 17:03:50.04 |
| | DRQN | 00:00:12.03 |
| | EULER-VI | 1:43:54.33 |
| Mountain Hike | ACNO-A2C | 1 day, 15:15:37 |
| | DVRL | 1 day, 20:30:01 |
| | Standard A2C | 4:38:59 |
| Cart Pole | ACNO-A2C | 1 day, 6:59:50 |
| | DVRL | 1 day, 13:50:49 |
| | Standard A2C | 1:42:38 |

## Appendix B.5 Full experimental results

**Table 3**  Table 3 in the manuscript shows the average discounted returns and 1 standard error from 3 separate seed averages (for example, for EULER-VI with observation cost of -0.1 the mean of the expected rewards is calculated from the results of {0.478, 0.484, 0.513} and the standard errors included in the table below indicate the errors over 50 simulated runs), each with rewards averaged across 50 simulated rollouts. The table below shows the average reward and 1 standard error over 50 simulation episodes at the end of training (each row shows the results from a different run).

| Observation Cost | *Observe-then-Plan* (Observe before planning) | ACNO-POMCP (Observe while planning) | DRQN (Generic POMDP-RL) | EULER-VI (MDP-RL) | POMCP-planning (With true models) |
|---|---|---|---|---|---|
| -0.1 | $0.739 \pm 0.023$ | $0.554 \pm 0.040$ | $0.650 \pm 0.020$ | $0.478 \pm 0.019$ | $0.747 \pm 0.025$ |
| | $0.775 \pm 0.025$ | $0.653 \pm 0.023$ | $0.693 \pm 0.0$ | $0.484 \pm 0.021$ | $0.760 \pm 0.026$ |
| | $0.749 \pm 0.024$ | $0.600 \pm 0.032$ | $0.436 \pm 0.022$ | $0.513 \pm 0.019$ | $0.760 \pm 0.024$ |
| -0.05 | $0.735 \pm 0.024$ | $0.605 \pm 0.030$ | $0.649 \pm 0.010$ | $0.594 \pm 0.020$ | $0.758 \pm 0.023$ |
| | $0.713 \pm 0.026$ | $0.671 \pm 0.012$ | $0.525 \pm 0.013$ | $0.679 \pm 0.023$ | $0.751 \pm 0.021$ |
| | $0.772 \pm 0.020$ | $0.597 \pm 0.033$ | $0.605 \pm 0.021$ | $0.642 \pm 0.021$ | $0.764 \pm 0.024$ |

**Table 4**  This table shows the average discounted returns and standard deviations from 3 seed averages for **1000 episodes** (in the case of DRQN, the results show the last 1000 episodes from training and planning since DRQN mixes model training with planning). These values correspond to the second half of the plot from Figure 1 during which the agent switched from exploring to planning. For clarification, the difference between Table 3 and Table 4 is that Table 3 only reports results from 50 evaluation episodes, during which, for example, the networks of DRQN were frozen while the results in Table 4 report the average returns from continuously training the model throughout the 2000 episodes (unless the algorithm switches from exploring to planning), which are also presented in Figure 1 of the main text.

Since we do not know the exact optimal behavior in the Sepsis environment, we provide the results from POMCP-planning with the true model parameters to show the planner's performance when model estimation is assumed to be optimal.

| Observation Cost | *Observe-then-Plan* (Observe before planning) | ACNO-POMCP (Observe while planning) | DRQN (Generic POMDP-RL) | EULER-VI (MDP-RL) | POMCP-planning (With true models) |
|---|---|---|---|---|---|
| | $0.743 \pm 0.174$ | $0.592 \pm 0.236$ | $0.524 \pm 0.156$ | $0.477 \pm 0.134$ | $0.725 \pm 0.180$ |
| -0.1 | $0.738 \pm 0.182$ | $0.609 \pm 0.227$ | $0.607 \pm 0.149$ | $0.512 \pm 0.142$ | $0.728 \pm 0.182$ |
| | $0.748 \pm 0.165$ | $0.606 \pm 0.232$ | $0.613 \pm 0.186$ | $0.505 \pm 0.150$ | $0.736 \pm 0.168$ |

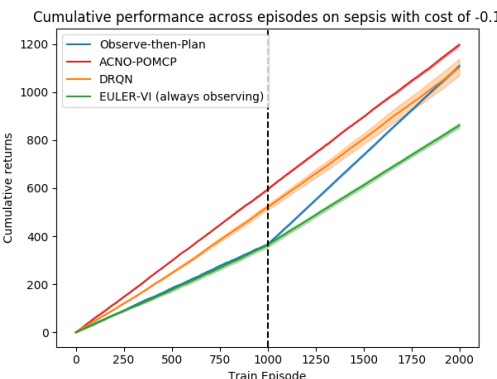

Figure 3: Supplement to Figure 1. Shows cumulative returns across 2000 episodes of model estimation and planning. The dotted vertical line shows when the algorithm switches from exploring purely for model estimation to planning for an optimal action sequence.