# OpenReview forum: "Reinforcement Learning with State Observation Costs in Action-Contingent Noiselessly Observable Markov Decision Processes "
_NeurIPS.cc/2021/Conference — NeurIPS 2021 Poster_

### Official Review · Reviewer_f13f · 2021-07-06

**Rating:** 7
**Confidence:** 3

**Summary:**

This paper concerns the class of POMDPs where the agent may obtain a perfect measurement about the hidden state at a cost. The paper argues such POMDPs, called ACNO-POMDPs, appear in many realistic domains such as healthcare. The paper proposes a probably approximately correct (PAC) learning algorithm for ACNO-POMDPs, and proves a finite sample complexity guarantee for it. The sample complexity guarantee is better than in general POMDPs. Further, the paper proposes an extension of the proposed ideas to a more realistic continuous-state and continuous-action setting. The performance of all proposed methods is empirically quantified in three experimental domains, showing that ACNO-POMDP specific methods provide improvements over general POMDP solvers.

**Limitations And Societal Impact:**

The limitations and societal impact are adequately addressed.

**Main Review:**

### Strengths

- The work here attempts to isolate the hardness in POMDP learning due to partial observability by considering a less general but still interesting case where the state is perfectly observable to the agent by paying a sensing cost. The paper coins the term ACNO-POMDP for such domains, and argues that they appear in many realistic applications. This is a novel angle and differs from other proposals in POMDPs dealing with sensing and information gathering, such as using belief-dependent rewards (Araya-Lopez et al., NIPS 2010).

- By consider the (very special) case of a costly perfect measurement channel, the paper is able to bring into play results from fully-observable MDPs which helps the analysis. This technical approach is new to me and it might have further uses in analyzing POMDPs.

- I lack the expertise to fully verify the proofs, but the techniques and overall results appear plausible. The sample complexity result is quite interesting, concretely showing that leveraging knowledge about the observation model leads to improvements over the general POMDP case.

- I also appreciate the extension of the ideas to the continuous-state and continuous-action case, which shows that the results here also have potential impact in large-scale problems.

- The paper is well written and easy to follow.


### Limitations

1. The idea of the sepsis domain experiment is sound, and it would be very useful to demonstrate benefits of ACNO-POMCP in a simple case, as in the other domains (Sect 6.2) there are many other potentially conflating factors. But, I am not sure if the sepsis domain is the best choice. Considering the tabular setting of Table 1, ACNO-POMCP performs within error bounds of generic DRQN, with no significant improvement. That "observe-then-plan" that uses much more exploration works better, which is not wholly surprising considering the algorithmic differences. But again, the sepsis domain is also one example of a high-stakes domain where such exploration might not be feasible, as appropriately acknowledged in the conclusion of the paper.

2. The theoretical results concern the "observe before planning" setting, which requires extensive exploration. This could be impossible in some domains (see also point above), which somewhat diminishes the potential impact of these results.

3. The experimental results reported could be a bit deeper. They do report good overall performance, but some additional ablation studies could shed further light into the POMCP-style or A2C-flavoured methods proposed. It would be interesting to see for example the effect of different belief encodings vs. using a Bayes filter with the learned models. Another interesting baseline might be a planning solution in the tabular domain, to determine the gap between an optimal solution and the "observe-then-plan" or ACNO-POMCP solutions. Reporting any qualitative differences between the policies found by the proposed method versus generic POMDP RL would help understand the methods better. To be clear, I do not demand that specifically these ablations be conducted, rather that any such experiments to provide additional insight would be valuable.


### Minor comments
- From Line 134 onwards, it seems ill-suited to define the policy as a mapping from states to distributions over actions. After all, in the ACNO-POMDP setting the agent might does not necessarily have access to the state. More appropriate would be to define the policy as mapping from action-observation histories or beliefs states.
- Reference numbering in the appendix does not seem to match the main paper, e.g., [62] vs. [63]. This should be corrected.

### After author reply
I am happy with the author response to my comments. I welcome the proposed changes to the manuscript and remain by my accept rating.

**Time Spent Reviewing:**

5

---

> ### Author Response · Authors · 2021-08-10
> **Response to Reviewer f13f**
>
> Thank you for the helpful and positive feedback and suggestions.
>
> 1. **The theoretical results of “observe before planning” require extensive exploration (may be unrealistic in some settings).**
>
> We completely agree that first learning the dynamics before planning will not be realistic in some settings. However, we believe that our work takes a useful step towards characterizing the provable benefits in sample efficiency in ACNO-MDP RL compared to generic tabular POMDP RL, and the approach of observing before planning, where exploration is conducted in the first phase, may be particularly useful when there is a finite budget for exploration. We are happy to expand on this in the text and also to highlight the interesting directions for future work on analyzing integrated approaches, noting that in generic continuous-state settings, theoretical analysis for sample complexity or regret analysis for POMDPs or even MDPs remains largely an important open area of research.
>
> 2. **Suggestion of expanded ablation studies and additional baseline of planning solution in the tabular domain**
>
> We thank the reviewer for the interesting additional ablation study ideas and we would be happy to add an additional comparison to a different belief encoding (in the Sepsis domain as well as a more simple case), include the optimal policy in the tabular domain to highlight an upper bound, and include a short description to compare the qualitative differences between our methods and generic POMDP RL.
>
> 3. **Policy as a mapping from states to distributions over actions (Line 134~) seems ill-suited and fix reference numbering in appendix**
>
> We appreciate the reviewers’ catching these small typos which we will fix. The policy should indeed be a mapping from belief states to actions, and this is the definition on which our analysis is correctly based.
>
> 4. > This technical approach is new to me and it might have further uses in analyzing POMDPs. I also appreciate the extension of the ideas to the continuous-state and continuous-action case, which shows that the results here also have potential impact in large-scale problems. The paper is well written and easy to follow.
>
> Thank you!

---

### Official Review · Reviewer_7nyK · 2021-07-14

**Rating:** 6
**Confidence:** 4

**Summary:**

This paper presents a framework to handle a class of problems where an agent, operating in an uncertain and partially observable environment, makes decisions that attempt to maximize a discounted finite-horizon reward. The framework assumes that the agent has some ability to fully infer the state of the system through a costly but concurrent action with the agent's planning function. The authors propose a novel RL algorithm whereby the agent initially learns elements of the dynamics and then applies the learned models to a POMDP planning framework. Additionally, they compare the proposed algorithms to modifications of existing RL algorithms in this unique setting.

**Limitations And Societal Impact:**

The healthcare example that the authors list is definitely a good fit with the approach and the formulation. They also do acknowledge the implications and limitations of performing online reinforcement learning in a hospital ICU, which is an important point.

**Main Review:**

--- after rebuttal: The authors addressed my main concerns, and clearly spent a strong effort on the rebuttal. I'm therefore raising my score.

Originality: POMDP reinforcement learning is a very diverse field with a disparate understanding of model-types and model classes. Certainly, the authors should not be penalized for not covering every potential instance of mixed-observability Markov decision process. However, they are certainly not the only work where the agent may select an action, at a cost, to reduce uncertainty in their belief of the environment. While they do not exactly match this unique formulation, the following are a few examples (there are many) where similar ideas are explored:

Mixed observability MDPs (MOMDP), in particular, RockSample:

[1] Smith, Trey, and Reid Simmons. "Heuristic search value iteration for POMDPs." arXiv preprint arXiv:1207.4166, 2012.

Active perception frameworks:

[2] Ghasemi, Mahsa, and Ufuk Topcu. "Perception-Aware Point-Based Value Iteration for Partially Observable Markov Decision Processes." IJCAI, 2018.

[3] Matthijs Spaan and Pedro ULima. A decision-theoretic approach to dynamic sensor selection in camera networks. In ICAPS, 2009

Also relevant:
Non-observable MDPs (NOMDP)

[4] Littlefield, Zakary, et al. "The importance of a suitable distance function in belief-space planning." Robotics Research. Springer, Cham, 2018. 683-700.

Quality:
The policy and optimal policy definitions as defined in lines 135-139 are for an MDP policy and its corresponding optimal policy, which do not hold for POMDPs. The optimal policy for a POMDP is the $\max_{\pi} V^{\pi}(b)$ for all beliefs $b$ in the belief space.

EEPORL [12], the structure upon which the authors build their observe before planning algorithm make a strong assumption about the POMDP domain. Assumption 2 - It is possible to achieve a non-zero probability of being in any state in two steps from the initial belief.
The examples used in this paper do not follow this assumption, which is quite strong.
The simplifying assumption in Definition 3 (line 655 of Appendix) is not sound. The $\alpha$-vector, which represents your t-step conditional policy, is not stationary in this instance since it depends on $\alpha^{\pi_{t-1}_{t-1}(s)$. Further the observation function is effectively eliminating non-observation actions from your analysis so the bound will only apply when an observable action is taken, which reverts to MDP-RL.

Clarity:
The authors are operating with significantly truncated and imprecise definitions of key elements in the framework. For instance, introducing the MDP first would ease the understanding of ``control actions''. Then, explaining how this fits into your particular POMDP, where the action space is modified to account for the observation, would be a more succinct method than introducing the concept of ``missingness''. While occasionally such truncation is required to effectively fit all the material they were attempting to cover into the 9 page limit, in this instance I believe they would have been better served shortening section 6 to provide additional clarity in section 3. Section 3 does not give enough context to conduct the analysis in Section 5 - attempting to provide both analysis and extended experiments in the one paper involves a truncation in both and does justice to neither.

Significance:
The work presents interesting directions for analysis, in particular, the ANCO-MDP model is an useful structure - albeit complicated to convey formally. However, the comparisons to existing learning approaches are still arbitrary (switching between observing and planning in the tabular settings) and problem specific to justify publication.

**Time Spent Reviewing:**

5.5

---

> ### Author Response · Authors · 2021-08-10
> **Response to Reviewer 7nyK**
>
> Thank you for your thoughtful and detailed comments! We will update the text to improve its clarity as per your suggestions. In the following we respond to your other questions and comments. Our understanding is that the majority of concerns are due to minor misunderstandings which we caused by some confusing wording: we apologize for this and we think a small set of edits that we will do will resolve the raised issues.
>
> 1. **Additional related work on POMDPs with observation costs.**
>
> We are happy to include these additional references to POMDP papers where observations can have costs. The papers mentioned are all POMDP planning papers where the dynamics and observation models are known; in contrast, our focus is POMDP RL, specifically ACNO-MDPs, where the agent does not know the dynamics model. We are happy to cite these and clarify the distinction.
>
> 2. > The policy and optimal policy definitions as defined in lines 135-139 are for an MDP policy and its corresponding optimal policy, which do not hold for POMDPs. The optimal policy for a POMDP is the $\max_{\pi} V^{\pi}(b)$ for all beliefs $b$ in the belief space.
>
> Thank you for catching this typo: the reviewer is completely correct that the value function for the POMDP is over the belief state. We will fix this typo. The later results and analysis (Section 5 and appendix) are all for the value function over belief state, as it should be.
>
> 3. **Use of EEPORL Assumption 2.**
>
> We apologize for our confusing statement in the proof sketch. While we build on the EEPORL analysis to show that it is possible to compute a near-optimal policy given a sufficiently accurately learned dynamics and observation model, our analysis and algorithm to ensure the dynamics models are sufficiently accurately learned is quite different than EEPORL and we do not require EEPORL’s assumption 2 [assumption 2 in that paper requires all states being reachable in 2 steps from the first step]. Our sample complexity analysis is for an algorithm which observes first, making this a fully observable MDP, and then continues to observe until the dynamics are learned sufficiently accurately such that we can compute a near optimal policy in the POMDP (using the learned dynamics and the known observation model). We will be sure to clarify this in the text.
>
> 4. > The simplifying assumption in Definition 3 (line 655 of Appendix) is not sound. The α-vector, which represents your t-step conditional policy, is not stationary in this instance since it depends on $\alpha^{\pi_{t-1}}_{t-1}(s')$.
>
> We apologize for the confusion. As stated briefly in section 2, we consider finite-horizon decision processes, so our alpha vectors will be non-stationary in general to account for the finite horizon episodic setting.
>
> 5. > Further, the observation function is effectively eliminating non-observation actions from your analysis so the bound will only apply when an observable action is taken, which reverts to MDP-RL.
>
> Thank you for the opportunity to clarify this. Actually our bound does hold for the full ACNO-MDP setting described in theorem 1. Note that in our setting we can view each action as having two versions: an observed or unobserved version. In both versions, the influence of the action on the state is identical: $p(s’|s,a)$ is the same for the unobserved version of $a$ and the observed version. Therefore the learned dynamics from the observed version of an action $a$ is identical to its unobserved counterpart. The only difference is that the unobserved action counterpart is associated with deterministically observing the “missingness” observation instead of deterministically observing the true state. We will be sure to make this clearer.
>
> 6. >  I believe they would have been better served shortening section 6 to provide additional clarity in section 3.
>
> We thank the reviewer for these helpful recommendations. We will apply this advice and restructure as needed in order to bring more of the precision and clarity regarding ACNO-MDP definitions and formalisms from the supplement into the main text. We will also restructure section 3 to introduce the MDP first before discussing control actions and “missingness” observations.
>
> 7. > However, the comparisons to existing learning approaches are still [too] arbitrary (switching between observing and planning in the tabular settings) and problem-specific.
>
> We appreciate the opportunity to address this. While we completely agree there are many potential approaches to learning in ACNO-MDPs, we do wish to highlight two things. First, our approach of learning the models (through observing) and then computing a near-optimal policy is motivated by the tradition of prior work in sequential decision making under uncertainty literature (like E^3 (Kearns and Singh), Epoch Greedy (Langford and Zhang)) and controls research that performs system identification to learn the domain models before computing an optimal policy. The second method is inspired by the practical success of recent POMDP-RL algorithms and demonstrates how such methods can be adapted to leverage the structure of ACNO-MDPs, and that such adaptation yields significant success.
>
> **References:**
> - Kearns, Michael, and Satinder Singh. "Near-optimal reinforcement learning in polynomial time." Machine learning 49.2 (2002): 209-232.
> - Langford, John, and Tong Zhang. "Epoch-Greedy algorithm for multi-armed bandits with side information." Advances in Neural Information Processing Systems (NIPS 2007) 20 (2007): 1.
>
> 8. > The work presents interesting directions for analysis, in particular, the ANCO-MDP model is a useful structure - albeit complicated to convey formally.
>
> Thank you!

---

### Official Review · Reviewer_GV21 · 2021-07-15

**Rating:** 6
**Confidence:** 3

**Summary:**

This paper proposes action-contingent noiselessly observable MDPs (ACNO-MDPs), where the agent has option to fully observe the latent state after each of their action albeit with a cost. ACNO-MDPs are a special case of POMDPs and the additional structure enables authors to show tighter bounds on the number of episodes exhibiting worse than near-optimal performance.

**Limitations And Societal Impact:**

Authors have adequately addressed the limitations and potential negative societal impact of their work.

**Main Review:**

Formulation of ACNO-MDPs is very novel and it models a highly important problem: how to balance the cost of collecting information and the returns gained through having additional information? The paper gives good examples of scenarios where this problem might arise.

However, the algorithms proposed to solve the ACNO-MDP problem lack novelty as they mostly rely on existing solutions to the general POMDP problem. The paper considers two frameworks: "observe before planning" and "observe while planning." While the observe-before-planning solution is EULER [62] followed by POMCP [49], the observe-while-planning solution is just POMCP with the added structure of ACNO-MDPs.

In the case of observe-before-planning, one novel question to answer could have been how to determine N optimally such that the cost of additional observation during the first N episodes is balanced with the performance gain in however more episodes the algorithm acts. However, the paper ignores the additional cost of the initial exploration phase completely and just bounds the number episodes needed to act near-optimally in the second phase (Theorem 1). Using this bound as N is not only impractical but also the bound only holds if the POMCP algorithm is replaced with other POMDP planning algorithm with optimality guarantees, which are also said to be impractical to use. (In the experiments, N is just set to be 1000 arbitrarily.)

In the case of observe-while-planning, the paper proposes a belief encoder, which is an adaptation of the DVRL [33] belief encoder. Highlighting how this adaptation differs from the DVRL version could improve the novelty of the proposed solutions.

Other minor comments:
* How realistic is the assumption of being able to fully observe the state in domains given as examples? For instance, in healthcare, performing a test only reveals one aspect of how healthy the patient is.
* In Table 2, how about the sample complexity of an MDP-based algorithm. Similar to how ACNO-MDP is a special case of POMDPs, it can also be viewed as a generalization of MDPs. Seeing how much more challenging learning in this environment is compared with a simple MDP would also be informative.
* In Table 3, the returns of the policy reached after 2000 training episodes is not a fair metric to compare the observe-then-plan solution with the observe-while-planning solution. This metric does not consider the total cost incurred during the training phase hence it favors the former solution, which ignores cost of observing in the first 1000 episodes. A better metric (or an additional metric to consider) could be regret.

**Time Spent Reviewing:**

4

---

> ### Author Response · Authors · 2021-08-10
> **Response to Reviewer GV21**
>
> We thank the reviewer for their thoughtful and detailed comments. They provide opportunities for clarification and improvement on which we elaborate in the points below.
>
> 1. > Formulation of ACNO-MDPs is very novel but the algorithms proposed to solve the ACNO-MDP problem lack novelty.
>
> We thank the reviewer for recognizing the novelty and impact of our ACNO-MDP formulation, which we view as a primary contribution of the paper. We agree that there are likely to be many interesting algorithms for learning in such domains. However, we view our initial simple approach as a strength, showing that we can already achieve substantially improved theoretical bounds and practical performance gains by adapting to prior work in order to leverage the structure of this setting. Note that doing so still requires care to a number of technical details to ensure we will learn a sufficiently accurate MDP model that can be used for obtaining a near-optimal policy in the original ACNO-MDP, and to bound the resulting sample complexity. As an additional contribution, to the best of our knowledge, our work is the very first to provide any sample complexity guarantees for this class of problems with rank-deficient observation matrices: prior POMDP-RL approaches assume that the observation probability matrix is full column-rank (with rows representing observations and columns representing states, each column summing to 1), but the observation probability matrix following a non-observing action in ACNO-MDPs has rank 1 because the agent receives a “missingness” observation with probability 1 regardless of the latent/true state.
>
> 2. > In the case of "observe-before-planning" [...] the paper ignores the additional cost of the initial exploration phase completely and just bounds the number episodes needed to act near-optimally in the second phase. [...] A better metric (or an additional metric to consider) could be regret.
>
> While we definitely agree that regret is an interesting metric to consider, in this paper we focus on obtaining a sample complexity result under the probably approximately correct (PAC) framework which has been extensively considered in MDPs. This can be particularly useful when there is a finite budget for exploration. We agree a regret analysis would be an interesting direction for future work. We will revise our text to report cumulative rewards (including observation cost) obtained in our experiments in the appendix. We will also include comparisons to the sample complexity of related MDP-based algorithms, as suggested.
>
> 3. > Using this bound as N is not only impractical but also the bound only holds if the POMCP algorithm is replaced with other POMDP planning algorithm with optimality guarantees, which are also said to be impractical to use.
>
> We completely agree that the theoretical results do not provide a practical sample complexity bound for learning the dynamics model. Unfortunately, this is common -- theoretical sample complexity bounds in the tabular MDP and POMDP RL literature suggest far more samples than is typically needed in empirical settings, suggesting an interesting opportunity for future work on instance-dependent bounds and structure. Such bounds are often still considered useful, both for helping characterize the hardness of settings and problems, and because when combined with a hyperparameter (e.g., the tolerability of accumulated negative rewards during exploration), they may be used to guide exploration in applications.
>
> Note that fortunately in tabular settings, it is often practical (using MCTS approaches etc) to compute a near-optimal POMDP policy by only focusing on extracting a decision for the current belief state.
>
> 4. > In the case of observe-while-planning, the paper proposes a belief encoder, which is an adaptation of the DVRL [33] belief encoder. Highlighting how this adaptation differs from the DVRL version could improve the novelty of the proposed solutions.
>
> We apologize for the lack of clarity here and we are happy to expand on this in the text. Briefly, our belief encoder is trained to directly estimate the next state, as the model weights are updated after every observation action. For observation actions, next states are not estimated but set to the received observations. For non-observation actions, next states are estimated by the current transition network, and the weights of the estimated state particles are assigned according to -log likelihood of p(next state estimates | actions & GRU-aggregated past history based on states, observations, and actions). This is different from the DVRL belief encoder which (1) always estimates the next state particle regardless of the non/observation actions, and (2) learns the observation function simultaneously with the transition network.  We have more details of each encoder in the appendix (L954~970 for the DVRL belief encoder and L971-983 for the proposed ACNO-A2C belief encoder).
>
> 5. > How realistic is the assumption of being able to fully observe especially in healthcare domains?
>
> We thank the reviewer for this question: we completely agree that it is not always appropriate to assume we can completely observe the state. However, in many settings the available tests may capture necessary details of the state and randomness in transitions are governed by inherent stochasticity in the system rather than an inability to observe the underlying state. For example, glucose monitoring to assist in insulin dosing recommendations and white blood cell count monitoring to assist in anti-HIV drug dosing seek to directly control certain patient biomarkers. In these cases, states can be more readily conceptualized as the relevant biomarkers themselves, making the applicability of the ACNO-MDP framework more clear. We will update the text to be clearer about this and provide these motivating examples.

---

### Official Review · Reviewer_pVS8 · 2021-07-25

**Rating:** 7
**Confidence:** 4

**Summary:**

The paper introduces a new specialisation of POMDPs in which actions are annotated with an option to perform additional sensing. Additional sensing incurs extra cost, but its observations are deterministic, which means that the planning agent receives full information about the underlying hidden state whenever the new sensing action is performed. This POMDP formulation is sensible, and there may be practical applications in which it could be used. The authors derive PAC bounds for planning with this new POMDP specialisation, and propose planning algorithms that can exploit the specific nature of their POMDP definition. Both the theory (i.e. PAC bounds) and the empirical results are encouraging.

**Limitations And Societal Impact:**

No need for improvement.

**Main Review:**

The technical contribution is strong, and the overall quality of writing is excellent and most parts of the paper are very clear and easy to follow. I did not try to verify  the proofs, which means that I might have missed potential problems in the purely formal lines of reasoning. I believe that this could be a good NeurIPS paper, but I could suggest a few improvements.

Perhaps the authors could provide some basic intuitive justification as to why the ACNO-MDPs provide tighter bounds. Is this because observations are deterministic and the agent can gain full information about the hidden state whenever it chooses to do so? I am sure that one could find many POMDP benchmarks in which entropy of the belief state is always high. Here, the agent can reduce this entropy to zero if needed. I would be interested in knowing if this the main reason for tighter bounds, or maybe I am missing something and there are other reasons. Having a clear discussion on this in the paper would be highly beneficial.

The user-adaptive experience example has some weaknesses. As long as it is true that constant sensing on mobile devices is energy consuming, the energy requirements of running a POMDP policy itself can be much higher when a POMDP is solved using Monte Carlo planning or a deep neural network is involved. Over the last 10 years multiple research groups have observed that finite-state controllers could be more appropriate for executing POMDP policies on mobile devices. I would say that combining ACNO-MDPs with finite state controllers or equivalent could make the user-experience example more realistic. I cannot buy it in this form.

On page 5, the authors said "While in our analysis we leverage algorithms that guarantee epsilon-optimal performance in POMDP planning [46, 10], these approaches are
often impractical for any reasonably-sized POMDP.". I think that this is a bold statement or it should be clarified. Does it mean that all the existing research on PAC-bounds is useless? Do the authors want to say that PAC bounds are impractical? This statement can be aggravating for some researchers.

The paragraph that is between lines 197-209 is a bit unclear for someone who does not know [33]. Could you please check it? For example, after reading the paragraph I am not sure why I see GRU units in algorithm 2. GRU was not mentioned in the description of the method in the text above the algorithm.

The paper mentions ethical concerns in line 362. The authors should explain what the reason for those concerns could be. I would assume that in contrast to what the authors said, refusing to use advanced technology to help people would be unethical.

Many acronyms in the references section are not capitalised, e.g., POMDP is always pomdp, Bayesian should start with a capital B, and the "US" shouldn't be "us". Every letter that should be capitalised could be simply put in curly braces in Bibtex. For example, if you type {B}ayesian in Bibtex, you will get Bayesian in your list of references in Latex.

Equations on pp. 22 and 23 have some formatting issues. Also, the table on p. 28 is too wide.

Time in line 1019 is in seconds whereas in line 1024 in days and hours. Please unify the time units. Note that tens of thousands of seconds are not easy to comprehend.

Assuming that the PAC proves are correct, this could be a nice NeurIPS contribution.

**Time Spent Reviewing:**

5

---

> ### Author Response · Authors · 2021-08-10
> **Response to Reviewer pVS8**
>
> 1. > The technical contribution is strong, and the overall quality of writing is excellent and most parts of the paper are very clear and easy to follow.
>
> Thank you! We appreciate the feedback. We're glad to hear that you similarly found the contribution strong and the writing easy to follow.
>
> 2. > Perhaps the authors could provide some basic intuitive justification as to why the ACNO-MDPs provide tighter bounds.
>
> Intuitively we can achieve tighter sample complexity bounds than generic POMDP RL because it is possible to directly observe the state. This allows our sample complexity guarantees to scale with the bounding error in the transition estimates, instead of as in generic POMDP RL which must detangle both environment stochasticity and observation uncertainty. We will be sure to include a discussion about this in the text.
>
> 3. > I would say that combining ACNO-MDPs with finite state controllers or equivalent could make the user-experience example more realistic.
>
> Thanks for bringing this point to our attention: we agree computational considerations are important and that leveraging finite state controllers for the POMDP planning part could be beneficial for the user-adaptive application. We will revise our text accordingly.
>
> 4. > The statement that “algorithms that guarantee epsilon-optimal performance in POMDP planning are often impractical for any reasonably-sized POMDP” should be clarified.
>
> We apologize for not being precise. Here we wish to simply highlight that computing an optimal policy for all belief states (doing exact POMDP planning) in a POMDP even with known parameters is computationally challenging in large domains, specifically PSPACE-complete (Papadimitriou and Tsitsiklis [1987]). However there are many advances in POMDP planning that allow near optimal policies to be computed for the visited belief state (including MCTS methods) and we will revise our text to reflect this.
>
> *Reference:*
>
> Papadimitriou, Christos H., and John N. Tsitsiklis. "The complexity of Markov decision processes." Mathematics of operations research 12.3 (1987): 441-450.
>
> 5. > The paragraph that is between lines 197-209 is a bit unclear for someone who does not know [33].
>
> We thank the reviewer for bringing this to our attention and will work to clarify this passage in the camera-ready version. Concretely, the GRU, or “belief encoder model” (line 204), is used in the belief encoder algorithm in order to generate compact representations of the agent’s state/action history (line 200), also referred to as the “particle’s [encoded] history” (line 202) or “history encoding”. The GRU “belief encoder model” is also used to estimate a particular state’s likelihood given an encoded history under the model. While one could replace the GRU in our algorithm with any similar kind of sequential model (e.g., LSTM), we recognize that it’s often easier to reason about concrete instantiations than abstract modules/tools. We will revise our text to clarify this passage.
>
> 6. > Some acronyms not capitalised, Equations on pg.22 and 23 have formatting issues, Table width on page 28, Time units in lines 1019 and 1024
>
> We appreciate the reviewer bringing these issues to our attention and we will fix them.
>
> 7. > The paper mentions ethical concerns in line 362. The authors should explain what the reason for those concerns could be.
>
> We thank the reviewer for highlighting this important point. Strategic exploration may require the agent to take actions that may not match best existing clinical guidelines, which may not be ethical in certain clinical settings. In addition, the policy learned optimizes for average performance but a risk-sensitive approach may be more appropriate in certain scenarios. We will expand this section to include this discussion.

---

> > ### Comment · Reviewer_pVS8 · 2021-08-16
> > **I've read your comments**
> >
> > Thank you for your time to answer reviewers' questions. I don't have any other queries at this point, and I am overall content with your explanations.

---

### Decision · Program_Chairs · 2021-09-27

**Decision:**

Accept (Poster)

**Comment:**

The paper proposes a new framework for partially observable RL where agents can pay an additional price to fuilly observe the environment.  This is a useful paradigm for several application domains.  The paper makes contributions both at the algorithmic and theoretical level.  Well done!